# Robust Policy Expansion for Offline-to-Online RL under Diverse Data Corruption

**Longxiang He**[1]   **Deheng Ye**[2]   **Junbo Tan**[1,†]   **Xueqian Wang**[1]   **Li Shen**[3,4,†]

[1]Tsinghua University    [2]Tencent    [3]Shenzhen Campus of Sun Yat-sen University

[4]Center for AI Theoretical Foundation and Systems, Shenzhen Loop Area Institute, China

longxhe@gmail.com    shenli6@mail.sysu.edu.cn

† Corresponding authors

## Abstract

Pretraining a policy on offline data followed by fine-tuning through online interactions, known as Offline-to-Online Reinforcement Learning (O2O RL), has emerged as a promising paradigm for real-world RL deployment. However, both offline datasets and online interactions in practical environments are often noisy or even maliciously corrupted, severely degrading the performance of O2O RL. Existing works primarily focus on mitigating the conservatism of offline policies via online exploration, while the robustness of O2O RL under data corruption, including states, actions, rewards, and dynamics, is still unexplored. In this work, we observe that data corruption induces heavy-tailed behavior in the policy, thereby substantially degrading the efficiency of online exploration. To address this issue, we incorporate Inverse Probability Weighted (IPW) into the online exploration policy to alleviate heavy-tailedness, and propose a novel, simple yet effective method termed **RPEX**: **R**obust **P**olicy **EX**pansion. Extensive experimental results on D4RL datasets demonstrate that RPEX achieves SOTA O2O performance across a wide range of data corruption scenarios. Code is available at https://github.com/felix-thu/RPEX.

## 1 Introduction

Deploying reinforcement learning (RL) in real-world environments without relying on simulation has long been a central goal in the field and is considered one of the most effective ways to realize the full potential of RL [45]. However, directly applying online RL [30, 11, 14, 15] in real-world settings is often expensive and unsafe due to the inherently exploratory nature of RL. To address this issue, Offline Reinforcement Learning (Offline RL), also known as Batch RL, has been proposed to learn optimal policies from pre-collected datasets without further interaction with the environment [12, 28]. Nonetheless, offline datasets typically lack full coverage of the state-action space, which often leads to suboptimal learned policies. Recently, inspired by the success of pretraining and fine-tuning paradigms [5, 9, 37], Offline-to-Online RL (O2O RL), which first trains a policy using offline data and then fine-tunes it through online interactions, has emerged as a promising paradigm for real-world learning [56, 36, 53].

Nevertheless, most O2O RL approaches [36, 56, 53] remain constrained to offline datasets derived from simulators, where data collection is relatively accurate. In contrast, real-world data—collected by humans or sensors and generated through online interactions—often contain unpredictable noise or even malicious corruption [50, 48, 51, 54]. For example, during fine-tuning data collection for Vision-Language-Action (VLA) models, human annotators may unintentionally or deliberately introduce incorrect trajectories into the dataset. Therefore, robust O2O policy learning that can handle data corruption in both offline and online phases is critical for the practical deployment of O2O RL.

39th Conference on Neural Information Processing Systems (NeurIPS 2025).

Most previous studies [43, 49, 4, 44, 38, 50, 48, 51, 54] on Robust RL have primarily focused on the theoretical properties and certification of offline reinforcement learning (RL) under data corruption. Notably, Yang et al. [50] highlights that existing offline RL algorithms, such as An et al. [2], suffer significant performance degradation when exposed to corrupted offline data. In contrast, IQL-style methods [25, 50, 17] demonstrate greater robustness to such corruption [50]. Nevertheless, as noted in RIQL [50], IQL remains sensitive to dynamics corruption, as data corruption induces a heavy-tailed distribution in Q targets. To address this, RIQL improves the robustness of IQL by incorporating the Huber loss [20, 21] and quantile-based Q estimators. Despite these advances, prior efforts have largely focused on robust offline RL. To the best of our knowledge, the challenge of learning robust and efficient O2O policies under both offline and online data corruption remains largely underexplored.

In this paper, we aim to develop a robust and efficient O2O method capable of handling data corruption occurring in both offline and online phases. Our key insight is that data corruption not only amplifies the heavy-tailedness of Q targets, as observed by Yang et al. [50], but also exacerbates the heavy-tailedness of the policy, which can severely hinder the efficiency of online exploration. Motivated by this observation, we propose a simple yet effective O2O approach, termed **RPEX** (**R**obust **P**olicy **EX**pansion), which integrates Inverse Probability Weighting (IPW)[42] into Policy Expansion[53] to alleviate policy heavy-tailedness. Experiments on D4RL tasks demonstrate that this simple modification yields an approximate 13.6% improvement in O2O performance over the vanilla PEX implementation.

To summarize, our main contributions are as follows:

- To the best of our knowledge, we are the first to propose a robust O2O reinforcement learning algorithm, RPEX, which achieves efficient and robust performance improvements under both offline and online data corruption.
- We demonstrate that data corruption not only amplifies the heavy-tailedness of Q-targets but also exacerbates the heavy-tailedness of the policy, thereby substantially impairing the efficiency of online exploration. Our theoretical analysis supports the effectiveness of our approach, demonstrating that applying IPW mitigates the heavy-tailedness induced by data corruption and facilitates efficient exploration.
- With a simple modification, extensive experimental results show that RPEX yields an additional performance improvement of approximately 13.6% over baseline methods.

## 2 Related Works

**Robust Offline RL.** Research on robust offline reinforcement learning (RL) can be broadly categorized into two areas. The first category [43, 49, 4, 44, 38] focuses on test-time robustness and sample complexity, aiming to learn from clean data while defending against adversarial attacks or noise during evaluation. The second category [50, 48, 51, 54] focuses on training-time robustness, which seeks to learn a reliable policy even from corrupted datasets, with evaluation conducted under clean conditions. Yang et al. [50] theoretically establishes the robustness of IQL [25] under various data corruption scenarios and introduces RIQL, which enhances IQL's robustness via Huber loss, observation normalization, and quantile Q estimators. Xu et al. [48] propose the Robust Decision Transformer to improve the robustness of DT under limited and corrupted data by incorporating embedding dropout, Gaussian-weighted learning, and iterative data correction. In contrast to prior work, we primarily focus on the Offline-to-Online (O2O) setting.

**O2O RL.** O2O reinforcement learning (RL) aims to enhance online sample efficiency while leveraging a pretrained offline policy. A central challenge in O2O RL is mitigating the conservative behavior induced during offline training to enable efficient exploration of high-quality samples. Naïvely continuing to train the offline policy with newly collected online data can be suboptimal and may degrade performance due to residual conservativeness from the offline phase. To address this, Nakamoto et al. [36] calibrates the offline value function to mitigate the "unlearning" or "forgetting" phenomenon. Zhou et al. [56] propose WSRL, which eliminates the need to retain offline data during online training. Zhang et al. [53] introduces policy expansion (PEX), which keeps the pretrained offline policy $\pi_\beta$ frozen and trains a separate learnable policy $\pi_\phi$ with independently initialized parameters, decoupled from $\pi_\beta$. This design retains the integrity of the offline policy while allowing the online policy to benefit from both prior knowledge and new interactions. Other approaches leverage Q-ensembles [55], data augmentation [31], or online RL exploration techniques [34] to

tackle similar issues. Our approach differs in that we explicitly address data corruption and noise during the O2O phase—an important but underexplored aspect in current O2O research.

The most relevant related works are Yang et al. [50] and Zhang et al. [53]. However, RIQL [50] focuses on robustness against offline data corruption, while PEX [53] focuses on addressing the unlearning phenomenon caused by directly transferring an offline policy to the online phase.

In contrast, our objective is to develop a robust O2O algorithm capable of withstanding various data corruptions in both offline datasets and online transitions.

## 3 Preliminaries and Problem Statement

**Offline RL.** Consider a Markov decision process (MDP): $M = \{\mathcal{S}, \mathcal{A}, P, R, \gamma, d_0\}$, with state space $S$, action space $\mathcal{A}$, environment dynamics $\mathcal{P}(s'|s, a) : S \times S \times \mathcal{A} \to [0, 1]$, reward function $R : S \times \mathcal{A} \to \mathbb{R}$, discount factor $\gamma \in [0, 1)$, policy $\pi(a|s) : \mathcal{S} \times \mathcal{A} \to [0, 1]$, and initial state distribution $d_0$. The action-value or Q-value of policy $\pi$ is defined as $Q^\pi(s_t, a_t) = \mathbb{E}_{a_{t+1}, a_{t+2}, \dots \sim \pi} \left[ \sum_{j=0}^\infty \gamma^j r(s_{t+j}, a_{t+j}) \right]$. The value function of policy $\pi$ is defined as $V^\pi(s) = \int_{\mathcal{A}} Q^\pi(s, a)\pi(a|s)da$. The goal of RL is to get a policy to maximize the cumulative discounted reward $J(\beta) = \int_{\mathcal{S}} d_0(s)V^\pi(s)ds$. $d^\pi(s) = \sum_{t=0}^\infty \gamma^t p_\pi(s_t = s)$ is the state visitation distribution induced by policy $\pi$ [45, 40], and $p_\pi(s_t = s)$ is the likelihood of the policy being in state $s$ after following $\pi$ for $t$ timesteps.

In an offline setting [12], environmental interaction is not allowed, and a static dataset $\mathcal{D}_{\text{offline}} = \left\{ (s_t, a_t, r, s'_{t+1}) \right\}_{t=1}^N$ is used to learn a policy. The key challenge in Offline RL is to avoid out-of-distribution (OOD) actions that arise when evaluating the learned policy [12, 28].

**O2O RL.** Given offline-pretrained value functions $Q_\phi, V_\psi$ and policy $\pi_\beta$, the objective of Offline-to-Online Reinforcement Learning is to obtain the optimal policy $\pi_\theta$ using the minimal number of online environment interactions.

**Data Corruption.** Following Yang et al. [50], Ye et al. [51], data corruption or attacks inject random or adversarial noise into the original states $s$, actions $a$, rewards $r$, and next states $s'$ drawn from either the offline buffer $\mathcal{D}_{\text{offline}}$ or the online buffer $\mathcal{D} = \left\{ (s_t, a_t, r, s'_{t+1}) \right\}_{t=1}^N$, resulting in corrupted buffers $\hat{\mathcal{D}}_{\text{offline}}$ and $\hat{\mathcal{D}} = \left\{ (\hat{s}_t, \hat{a}_t, \hat{r}, \hat{s}'_{t+1}) \right\}_{t=1}^N$. For example, in a random dynamics attack, random noise is injected into the next states, yielding $\hat{s}' = s' + \lambda \cdot \text{std}(s')$, where $\lambda \sim \text{Uniform}[-\epsilon, \epsilon]^{d_s}$, $d_s$ denotes the dimensionality of the state space, $\epsilon$ is the corruption scale, and $\text{std}(s')$ represents the $d_s$-dimensional standard deviation of all next states in the dataset. In contrast, adversarial corruption employs Projected Gradient Descent to optimize a noise $\epsilon$ for $\min_{\hat{s}} Q(\hat{s}, a)$ within a predefined region, using pretrained value functions. Further details on data corruption are provided in Appendix C.1. Throughout this paper, we slightly abuse the terms "corrupted" or "attacked" to refer to datasets or online trajectories that have been subjected to corruption.

**Implicit Q-learning (IQL).** To avoid OOD actions in offline RL, IQL [26] uses the state conditional upper expectile of action-value function $Q(s, a)$ to estimate the value function $V(s)$, which avoid directly querying a Q-function with unseen action. For a parameterized critic $Q_\phi(s, a)$, target critic $Q_{\hat{\phi}}(s, a)$, and value network $V_\psi(s)$ the value objective is learned by

$$\mathcal{L}_V(\psi) = \mathbb{E}_{(s,a) \sim \mathcal{D}}[L_2^\tau(Q_{\hat{\phi}}(s, a) - V_\psi(s))]$$
$$\text{where} \quad L_2^\tau(u) = |\tau - \mathbb{1}(u < 0)|u^2, \tag{1}$$

where $\mathbb{1}$ is the indicator function. Then, the Q-function is learned by minimizing the MSE loss

$$\mathcal{L}_Q(\phi) = \mathbb{E}_{(s,a,s') \sim \mathcal{D}}[(r(s, a) + \gamma V_\psi(s') - Q_\phi(s, a))^2]. \tag{2}$$

For policy extraction, IQL uses AWR [41, 40, 35], which trains the policy through weighted regression by minimizing $\mathcal{L}_\pi(\beta)$

$$\mathbb{E}_{(s,a) \sim \mathcal{D}}[-\exp(\alpha(Q_{\hat{\phi}}(s, a) - V_\psi(s)))\log \pi_\beta(a|s)]. \tag{3}$$

**Robust IQL (RIQL).** To enhance the robustness of IQL, particularly its resistance to dynamic attacks, RIQL [50] replaces the quadratic loss in Eq. (2) with the Huber loss $l_H^\delta(\cdot)$ and derives the following

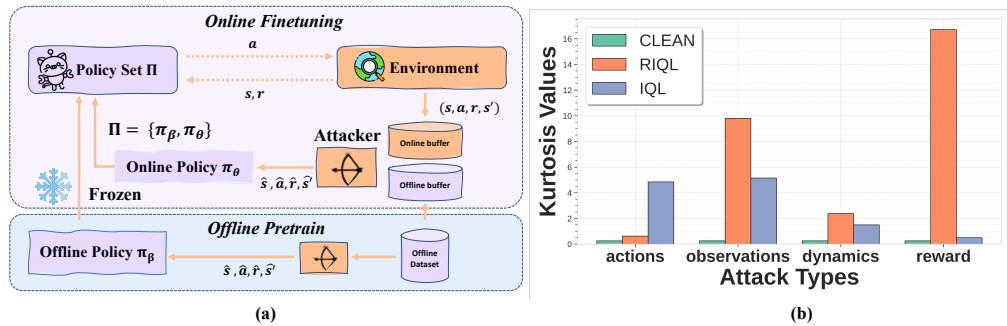

Figure 1: (a) Problem Statement. A schematic illustration of the O2O attack, in which both the offline pre-training phase and the online fine-tuning phase are targeted. (b) The Kurtosis Values [13, 33] of Policies. CLEAN means IQL is trained without attacks. In contrast, RIQL and IQL are trained on the attacked datasets.

loss function.

$$\mathcal{L}_Q = \mathbb{E}_{(\boldsymbol{s},\boldsymbol{a},r,\boldsymbol{s}')\sim\mathcal{D}}\left[l_H^\delta(r + \gamma V(\boldsymbol{s}') - Q(\boldsymbol{s},\boldsymbol{a}))\right], \quad \text{where } l_H^\delta(x) = \begin{cases} \frac{1}{2\delta}x^2, & \text{if } |x| \leq \delta \\ |x| - \frac{1}{2}\delta, & \text{if } |x| > \delta \end{cases},$$

(4)

where $\delta$ trades off the resilience to outliers from $\ell_1$ loss and the rapid convergence of $\ell_2$ loss.

**Policy Expansion.** To mitigate performance degradation during the transition from offline to online learning, PEX [53] utilizes a composite policy set $\Pi = [\pi_1, \ldots, \pi_K]$ and selects actions generated by the policies in $\Pi$ based on their potential utilities (e.g., critic values) in both exploration and policy learning. Specifically, for each element $\mathbb{A} = \{\boldsymbol{a}_i \sim \pi_i(\boldsymbol{s})\}$ in the policy set $\Pi$, assuming the size of $\Pi$ is $K$, the probability of selecting $\boldsymbol{a}_i$ as the final action is

$$P_{\mathbf{w}}[i] = \frac{\exp(Q_\phi(\boldsymbol{s},\boldsymbol{a}_i)/\alpha)}{\sum_j \exp(Q_\phi(\boldsymbol{s},\boldsymbol{a}_j)/\alpha)}, \quad \forall i \in [1,\cdots,K],$$

(5)

where $Q_\theta$ denotes the offline pretrained critic function, and $\alpha$ represents the temperature parameter. In both PEX and our method, the policy set is defined as $\Pi = [\pi_\beta, \pi_\theta]$ with $K = 2$, where $\pi_\beta$ is the offline pretrained policy and $\pi_\theta$ is the online learnable policy.

**Problem Statement.** In contrast to test-time robust RL [50, 48, 51, 54], which primarily emphasizes testing-time robustness—learning from clean data while withstanding attacks during evaluation—our setting centers on learning from corrupted data and evaluating in a clean environment [50, 48]. We assume access to an offline pretrained policy $\pi_\beta$ and value functions $Q(\boldsymbol{s},\boldsymbol{a})$ and $V(\boldsymbol{s})$, all trained on a corrupted offline dataset comprising states, actions, rewards, and dynamics. Our objective is to obtain an optimal policy based on $\pi_\beta$ through a limited number of online interactions. Furthermore, attacks or perturbations targeting states, actions, rewards, and dynamics are also present during online interactions. We model these attacks as affecting the online replay buffer, i.e., although the agent's interaction with the environment remains clean, data corruption occurs during storage in the buffer, leading to the use of compromised corrupted data for online policy updates. As we discussed in the Introduction, such scenarios are common and critical to the application of O2O RL in many real-world contexts [50, 48, 47, 51].

The complete problem statement is illustrated in Figure 1(a).

## 4 How does Heavy-tailedness of Policy Affect Performance?

As noted in RIQL [50], IQL exhibits notable robustness to data corruption in states, actions, and rewards, yet it remains vulnerable to dynamics attacks due to the emergence of heavy-tailed Q target, i.e., $r(\boldsymbol{s},\boldsymbol{a}) + \gamma V(\boldsymbol{s}')$. To address this, RIQL incorporates the Huber loss into the critic updates (Eq. (4)) to mitigate the heavy-tailedness of the Q target. However, we argue that the policy itself also becomes heavy-tailed as a result of such attacks.

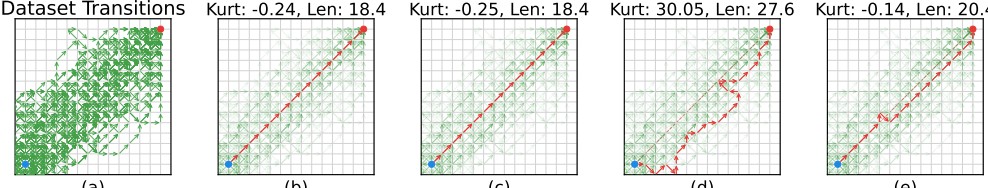

Figure 2: We study the impact of policy heavy-tailedness in the grid-world domain. An offline policy is trained using the dataset shown in Figure 2(a) and is then used to collect trajectories during the online exploration phase under both corrupted and uncorrupted settings. In Figures 2 (b)–(e), the opacity of the green arrows indicates the selection probability. Red arrows denote the most probable trajectory generated by IQL or IQL+IPW under the respective conditions. Specifically, panel (a) illustrates the dataset transitions; panels (b) and (d) show trajectories selected by IQL under clean and corrupted value functions, respectively; panels (c) and (e) show trajectories selected by IQL+IPW under clean and corrupted value functions, respectively.

To illustrate this, we visualize the **Kurtosis value**[1] of the offline policy $\pi_\beta(\boldsymbol{a}|\boldsymbol{s})$ pretrained by IQL or RIQL on the hopper-medium-replay task under various types of random attacks. We compute the Kurtosis value using 5000 samples collected after training IQL or RIQL for $2 \times 10^6$ steps. As shown in Figure 1(b), corruption-induced heavy-tailedness is more pronounced in the policy than in the Q target, whose heavy-tailedness primarily arises under dynamics attacks. Notably, the Kurtosis value of the IQL policy without attacks is very low (approximately $0.3$), compared to policies trained under corrupted conditions. Furthermore, as illustrated in Figure 1 (b), RIQL exhibits more severe policy heavy-tailedness than IQL, suggesting that RIQL alleviates the heavy-tailedness of the Q-function at the cost of increased heavy-tailedness in the policy.

During the offline phase, the policy is evaluated deterministically by selecting the action with the highest probability in the evaluation environments, which limits the influence of policy heavy-tailedness on offline performance. However, in the fine-tuning phase, the offline policy is utilized for exploration to further refine the pretrained policy. In this context, the heavy-tailedness hinders efficient exploration. We analyze this issue in detail through a series of simple toy maze experiments presented below.

Figure 2 illustrates the effect of policy heavy-tailedness during the online phase through a toy experiment, with experimental details provided in Appendix C.1. As shown in Figure 2(d), IQL samples poor-quality trajectories (i.e., longer in length) during exploration due to corruption-induced heavy-tailedness, as evidenced by elevated Kurtosis values.

**Key Observation:** Attacks on various components induce heavy-tailed behavior in the policy, leading to inefficient exploration.

Intuitively, the heavy-tailedness resembles common exploration mechanisms in policy learning, such as entropy maximization or injected exploration noise, as both assign small probabilities to certain actions. However, in contrast to intentional exploration strategies, the heavy-tailedness arises from malicious attacks and, more critically, introduces stochasticity that is beyond user control. In a standard offline-to-online setting, exploration is encouraged by assigning nonzero probabilities to suboptimal actions, with the degree of exploration regulated by predefined hyperparameters. In the presence of attacks, however, the heavy-tailedness in the policy assigns nonzero weights to inferior actions in an uncontrolled manner.

## 5 RPEX: Robust Policy Expansion

This observation motivates the adoption of Inverse Probability Weighting (IPW), a technique commonly used to address heavy-tailedness in classification tasks [42, 7, 52]. Specifically, given the offline pretrained $Q$, $V$, and a set of candidate policies $\Pi = [\pi_1, \ldots, \pi_K]$, we incorporate IPW into

---

[1]The Kurtosis value $\text{Kurt}[X] = \mathbb{E}\left[\left(\frac{X-\mu}{\sigma}\right)^4\right]$ quantifies heavy-tailedness relative to a normal distribution. For a detailed introduction, refer to Appendix A.

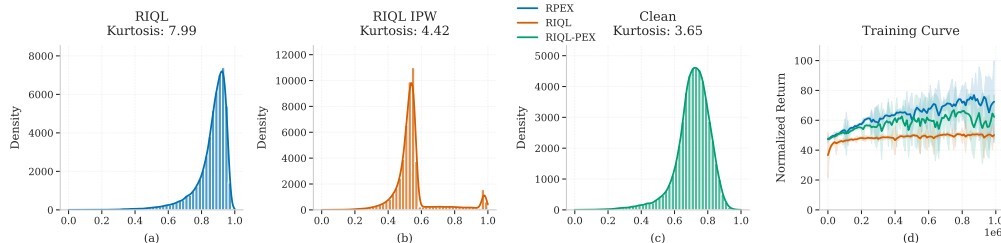

Figure 3: Action distributions generated by the offline pretrained policy under reward attack on the Halfcheetah-MR task. (a) Action distributions of RIQL under attack. (b) Action distributions of RIQL+IPW under attack. (c) Action distributions of IQL without attack. (d) Comparison of RPEX (with IPW) against RIQL-PEX (without IPW) and RIQL (Vanilla RIQL).

Eq. (5) to derive the following objective:

$$P_{\mathbf{w}}[i] = \frac{\exp(Q_\phi(\boldsymbol{s}, \boldsymbol{a}_i)/\alpha) + \kappa w_{\text{ipw}}^{\pi_i}}{\sum_j \exp(Q_\phi(\boldsymbol{s}, \boldsymbol{a}_j)/\alpha) + \kappa w_{\text{ipw}}^{\pi_j}}, \quad \forall i \in [1, \cdots, K], \tag{6}$$

where

$$w_{\text{ipw}}^{\pi_i} = \text{CLIP}\left(\frac{Q_\phi - V_\psi}{\pi_i(\boldsymbol{a}_i|\boldsymbol{s})}, \text{MIN}, \text{MAX}\right). \tag{7}$$

Here, $\alpha$ and $\kappa$ are coefficients, and $P_{\mathbf{w}}[i]$ denotes the probability of selecting $\boldsymbol{a}_i$. Throughout this paper, we clip the value of $w_{\text{ipw}}^{\pi_i}$ and set $|\text{MIN}| = |-10000| \gg |\text{MAX}| = |100|$ to primarily penalize low-quality actions while mitigating the impact of positive outliers.

**In what way does Eq. (6) enhance exploration efficiency?** Here, we provide an intuitive explanation of the behavior of Eq. (6). For low-quality actions ($Q - V < 0$), which generally have low selection probabilities and are widely distributed, the weight in Eq. (6) is predominantly determined by $w_{\text{ipw}}^{\pi_i}$. In such cases, low-quality actions that are rarely chosen offline become even less likely to be selected during online interaction. Conversely, for high-quality actions ($Q - V > 0$), the clipped IPW values cause the weights in Eq. (6) to be primarily influenced by $\exp(Q(\boldsymbol{s}, \boldsymbol{a}_i)/\alpha)$. This analysis indicates that Eq. (6) adaptively allocates weights according to action quality and eliminates the need for intricate confidence interval design, thereby mitigating the exploration inefficiency induced by long-tail behavior.

As illustrated in Figure 3 (a)-(c), data corruption can lead to heavy-tailedness in the policy and degrade O2O performance due to inefficient exploration (Figure 3 (d), Vanilla RIQL). Integrating IPW as presented in Eq. (6) can significantly enhance robustness against reward attacks (Figure 3 (d)), as heavy-tailedness is more pronounced in reward attacks, as shown in Figure 1 (b). Besides, in Section 6, we present a theoretical justification for the use of IPW.

In practical implementation, following Zhang et al. [53], we freeze the offline policy and set $K = 2$. We also examine the impact of commonly used techniques, such as state normalization [50] and the updates-to-data (UTD) ratio [56, 3], which denotes the number of updates performed per online trajectory, under O2O corruption settings. Notably, we find that state normalization improves the performance of O2O methods, whereas a high UTD ratio degrades performance under O2O corruption, except in the case of action corruption. Furthermore, policy extraction methods [17, 39, 16] play a critical role under state attack scenarios, extracting policies using AlignIQL [17] (Eq. (8)) yields superior performance under state attacks.

$$\pi^\star(\boldsymbol{a}|\boldsymbol{s}) \propto \mu(\boldsymbol{a}|\boldsymbol{s}) \exp\left\{-\eta\left(Q(\boldsymbol{s}, \boldsymbol{a}) - V(\boldsymbol{s})\right)^2\right\}. \tag{8}$$

A detailed ablation study investigating the impact of normalization, UTD, and policy extraction is provided in Sections 7.2 and Appendix C.2.

The overall RPEX algorithm is summarized below, with its pseudocode provided in Algorithm 1. First, RPEX inherits the offline pretrained policy, obtained via IQL or RIQL on the attacked offline dataset, and initializes a new online policy. It then fixes the offline pretrained policy and employs

Eq. (6) to collect online samples during the attacked online phase. Finally, RPEX trains the new online policy using a buffer that integrates offline and online data via IQL, RIQL, or other offline or online learning methods.

---

**Algorithm 1** RPEX: **R**obust **P**olicy **EX**pansion

---

**Input:** offline RL algorithm IQL or RIQL $\{L_{\text{offline}}^{Q_\phi}, L_{\text{offline}}^{\pi_\beta}\}$, online RL algorithm $\{L_{\text{online}}^{Q_\phi}, L_{\text{online}}^{\pi_\theta}\}^2$

**Initialize:** UTD$= M$, network parameters $\phi$, $\beta$, $\theta$, corrupted offline replay buffer $\hat{\mathcal{D}}_{\text{offline}}$

Normalize the states in both the environment and the corrupted offline replay buffer $\hat{\mathcal{D}}_{\text{offline}}$

**while** in *offline training phase* **do**

   % offline policy training using batches from the corrupted offline replay buffer $\hat{\mathcal{D}}_{\text{offline}}$

   $\phi \leftarrow \phi - \lambda_Q \nabla_\phi L_{\text{offline}}^Q(\phi), \quad \beta \leftarrow \beta - \lambda_\pi \nabla_\beta L_{\text{offline}}^{\pi_\beta}(\beta)$

**end while**

Policy Expansion: $\tilde{\pi} = [\pi_\beta, \pi_\theta]$; transfer $Q_\phi$

**while** in *online training phase* **do**

   **for** each environment step **do**

      $\boldsymbol{a}_t \sim \tilde{\pi}(\boldsymbol{a}_t|\boldsymbol{s}_t)$ according to (Eq. (6)),

      $\boldsymbol{s}_{t+1} \sim \mathcal{P}(\boldsymbol{s}_{t+1}|\boldsymbol{s}_t, \boldsymbol{a}_t),$

      Attack $\{(\boldsymbol{s}_t, \boldsymbol{a}_t, r, \boldsymbol{s}_{t+1})\}$

      % Add corrupted transition into online buffer $\hat{\mathcal{D}}$

      $\hat{\mathcal{D}} \leftarrow \hat{\mathcal{D}} \cup \{(\hat{\boldsymbol{s}}_t, \hat{\boldsymbol{a}}_t, \hat{r}, \hat{\boldsymbol{s}}_{t+1})\}$

   **end for**

   **for** each gradient step **do**

      % online training using batches from both $\mathcal{D}_{\text{offline}}$ and $\mathcal{D}$

      $\phi \leftarrow \phi - \lambda_Q \nabla_\phi L_{\text{online}}^Q(\phi), \quad \theta \leftarrow \theta - \lambda_\pi \nabla_\theta L_{\text{online}}^{\pi_\theta}(\theta)$ for $M$ times

      % high UTD for action corruption

   **end for**

**end while**

---

## 6 Theoretical Analysis

**Proposition 6.1.** *Given $P_{\mathbf{w}}(\boldsymbol{a}_i|\boldsymbol{a}_1, \boldsymbol{a}_2)$, where $\boldsymbol{a}_1 \sim \pi_\beta, \boldsymbol{a}_2 \sim \pi_\theta$, Eq. (6) maximizes the following objective (See proof in Appendix B.1.)*

$$\mathbb{E}_{\substack{a_1 \sim \pi_\beta, \\ a_2 \sim \pi_\theta}} \left[ \sum_i \underbrace{P_{\mathbf{w}}(i|s, a_1, a_2)\left(Q_\phi(s, a_i) - \alpha \log P_{\mathbf{w}}(i|s, a_1, a_2)\right)}_{\textit{Max Reward \& Entropy}} + \underbrace{\kappa_1 \frac{Q_\phi - V}{\pi_i(a_i|s)} P_{\mathbf{w}}(i|s, a_1, a_2)}_{\textit{Regularization}} \right].$$

(9)

The first term corresponds to the standard objective in maximum entropy RL [14, 15]. For the regularization term, we provide the following justification.

**Justification of Regularization.** This regularization term can be interpreted as a rectification mechanism. As discussed in Section 5, the heavy-tailed distributions induced by attacks increase the likelihood of assigning nonzero probabilities to "bad" actions, characterized by $Q(s, a) - V(s) < 0$. The regularization term penalizes such actions—e.g., $a_1$ when $Q - V < 0$—by reducing the probability of their selection, i.e., decreasing $P_{\mathbf{w}}(a_1|s, a_1, a_2)$. The magnitude of this penalization is modulated by $\kappa_1$ and $\pi_i(a_i|s)$; specifically, the smaller $\pi_i(a_i|s)$ is, the stronger the penalization, since a low action probability coupled with a poor $Q$ value ($Q - V < 0$) often results from corruption. Conversely, good actions ($Q - V > 0$) are encouraged, but controlled by our clip term in Eq. (7) to avoid placing too many weights on suboptimal actions.

---

$^2 L^Q, L^\pi$ denote the loss functions of the critic and actor, respectively, in actor-critic-based reinforcement learning. In our implementation, both the offline and online reinforcement learning algorithms are either IQL or RIQL.

Table 1: Average normalized Offline-to-Online score under random data corruption on the Medium-Replay Tasks over 5 random seeds.

| Environment | Attack Element | IQL | IQL-PEX | **IQL-RPEX (ours)** | RIQL | RIQL-PEX | **RPEX (ours)** |
|---|---|---|---|---|---|---|---|
| Halfcheetah-MR | observation | 21.4→21.7±5.8 | 21.4→21.5±2.1 | 21.4→**21.9**±2.8 | 19.73→21.3±4.2 | 19.73→20.9±5.3 | 19.73→**22.5**±3.1 |
| | action | 42.9→48.4±0.3 | 42.9→65.9±1.4 | 42.9→**69.2**±0.9 | 43.5→49.9±0.5 | 43.5→70.2±8.0 | 43.5→**77.8**±4.5 |
| | reward | 41.9→44.5±1.4 | 41.9→47.0±0.7 | 41.9→**52.7**±0.5 | 43.6→49.7±2.0 | 43.6→67.1±6.3 | 43.6→**73.6**±4.0 |
| | dynamics | 37.1→35.8±1.1 | 37.1→**37.0**±4.9 | 37.1→36.9±2.2 | 42.0→45.3±0.8 | 42.0→**44.7**±0.3 | 42.0→44.4±0.5 |
| Walker2d-MR | observation | 8.7→17.0±6.5 | 8.7→20.8±4.5 | 8.7→**23.3**±3.3 | 32.2→17.0±2.2 | 32.2→25.6±3.2 | 32.2→**30.5**±3.6 |
| | action | 64.7→**106.8**±0.9 | 64.7→105.3±0.6 | 64.7→106.4±0.5 | 85.9→48.8±20.9 | 85.9→109.2±15.6 | 85.9→**118.9**±10.1 |
| | reward | 77.2→90.1±9.9 | 77.2→90.1 ±7.2 | 77.2→**94.5**±6.5 | 81.8→91.3±1.6 | 81.8→91.9±1.1 | 81.8→**100.5**±2.9 |
| | dynamics | 14.9→4.5±1.8 | 14.9→**6.8**±2.5 | 14.9→4.7±1.2 | 80.0→87.4±2.7 | 80.0→89.5±1.3 | 80.0→**92.2**±1.4 |
| Hopper-MR | observation | 75.8→36.1±11.9 | 75.8→45.4±6.3 | 75.8→**76.9**±5.9 | 78.3→29.2±6.8 | 78.3→45.9±12.2 | 78.3→**73.1**±9.7 |
| | action | 93.4→95.0±4.4 | 93.4→102.2±6.4 | 93.4→**106.2**±5.7 | 75.9→95.8±3.2 | 75.9→93.2±10.5 | 75.9→**112.6**±5.4 |
| | reward | 55.2→97.8±1.2 | 55.2→99.8 ±2.4 | 55.2→**102.6**±2.8 | 72.9→68.3±2.9 | 72.9→90.1±2.9 | 72.9→**100.6**±2.4 |
| | dynamics | 0.8→6.0±7.5 | 0.8→0.7 ±0.0 | 0.8→**13.4**±0.9 | 44.6→54.2±13.6 | 44.6→51.0±4.5 | 44.6→**55.2**±4.7 |
| Average offline score ↑ | | 44.5 | 44.5 | 44.5 | 58.4 | 58.4 | 58.4 |
| Average O2O score ↑ | | 50.3 | 53.5 | **59.1** | 54.9 | 66.6 | **75.16** |
| Average improvement percentage ↑ | | 13.1% | 20.3% | **32.7%** | -6.1% | 14.1% | **28.7%** |

Overall, this regularization term supports our explanation in Section 5 regarding how Eq. (6) enhances exploration efficiency.

# 7 Experiments

In this section, we empirically assess the Offline-to-Online performance of RPEX under various data corruption scenarios.

**Experimental Setup.** Following RIQL [50], we use the corruption rates $c_1, c_2 \in [0, 1]$ and the corruption scale $\epsilon$ to control the overall level of data corruption. The offline pre-trained policy is obtained by injecting random noise into the corrupted elements of a $c_1$ fraction of the offline dataset. In the online phase, each trajectory is corrupted with probability $c_2$ by either random or adversarial noise. To better reflect real-world conditions, we adopt distinct corruption rates for the offline and online phases. In the main experiments, the offline corruption rate is set to $c_1 = 0.3$, the online corruption rate to $c_2 = 0.5$, and the corruption scale to $\epsilon = 1$ for both phases. For the Offline-to-Online score in Table 1, we report the mean results and standard deviation based on the final three evaluations averaged over five random seeds. Additional experimental details and implementation details are provided in Appendix C.1.

**Baselines.** For the offline pre-trained policy, we train IQL [25] and RIQL [50] with 2e6 gradient steps under randomly corrupted elements. We adopt IQL and RIQL as offline pre-trained policies because IQL is one of the most widely used offline RL baselines and has been shown to be robust to various data corruption scenarios [50], while RIQL is a robust variant of IQL. For the Offline-to-Online algorithms, we compare IQL, PEX [53], IQL+RPEX (ours), RIQL, RIQL+PEX, and RIQL+PREX (referred to as RPEX, ours). IQL and RIQL denote directly applying the respective algorithms in the Offline-to-Online setting without modification. Additional details about the baselines are provided in Appendix C.1.

## 7.1 Main Results

**Random Corruption.** The main results are obtained on the Medium-Replay tasks [10], which were collected during the training phase of a SAC agent and more closely resemble real-world conditions. Table 1 presents the average normalized performance of various algorithms. Overall, RPEX yields greater performance improvements compared to Vanilla IQL (RIQL) and IQL+PEX (RIQL+PEX), achieving average score gains of 32.7% over IQL and 28.7% over RIQL. In 11 out of 12 settings involving RIQL-based methods, RPEX achieves the highest performance, particularly under state, action, and reward corruption, where it significantly outperforms the baselines.

Additional experiments on mixed attacks, adversarial attacks, and AntMaze tasks are presented in the Appendix C.3.

## 7.2 Ablation Study

**UTD.** The updates-to-data (UTD) ratio refers to the number of updates performed per online trajectory. A high UTD generally accelerates policy learning in uncorrupted Offline-to-Online scenarios [56].

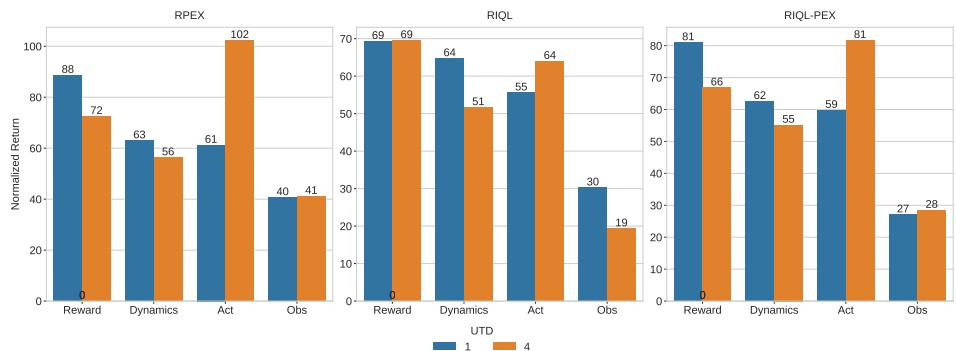

Figure 4: The effect of UTD for O2O methods. The results are averaged over the 5 random seeds on the Medium Replay tasks.

However, under corruption, a high UTD primarily improves robustness against action corruption and slightly degrades overall performance. This is likely because the policy update in IQL-style methods resembles supervised learning—Eq. (3) is akin to a cross-entropy loss with softmax. Such an update mechanism inherits the robustness properties of supervised learning [50, 48]. Consequently, a high UTD benefits from more gradient steps and mitigates degradation caused by noise under action corruption. We also conduct additional experiments in Appendix C.2, where a high UTD is applied only to the critic while the actor is kept at the standard setting (UTD=1), to further validate this phenomenon.

**Policy Type & Normalization & hyperparameter** $\kappa$ As noted in Yang et al. [50], IQL or RIQL with a deterministic policy and state normalization generally achieves better performance under data corruption. In this section, we examine the effects of policy type (deterministic or stochastic) and normalization during the O2O period using Hopper MR under action and dynamics attacks.

As shown in Figure 5, unlike in the offline setting, a deterministic policy can degrade O2O performance. This is because exploration in deterministic policies is typically achieved by adding small noise to actions, which may lead to inefficient exploration. In contrast, applying state normalization significantly enhances O2O performance. Furthermore, the ablation study on $\kappa$ in Figure 5 indicates that RPEX is relatively insensitive to $\kappa$, although fine-tuning this parameter can still yield marginal improvements. Note that all results in Table 1 are reported using $\kappa = 0.1$ and inverse temperature $\alpha^{-1} = 3$. A detailed ablation study on offline buffer retention, clipping range, and the policy extraction method is presented in Appendix C.2. In addition, Appendix C.3 reports the training cost of our method and the evaluation of RPEX under varying levels of data corruption.

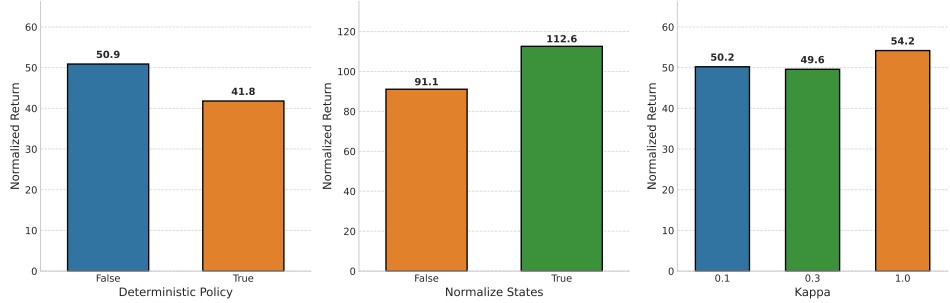

Figure 5: Ablation study of the main components of RPEX on Hopper MR task. From left to right: policy type ablation, state normalization ablation, and hyperparameter $\kappa$ ablation.

## 8 Conclusion and Limitations

In this paper, we propose RPEX, a method designed to enable robust online fine-tuning under O2O data corruption by leveraging importance sampling (IPW). This setting is critical for deploying

reinforcement learning in real-world environments, where noise and adversarial attacks are ubiquitous. RPEX integrates IPW into PEX to mitigate the heavy-tailed behavior of policies induced by corrupted data. Through empirical evaluations under random, adversarial, and mixed attacks, we demonstrate the robustness of RPEX against various forms of data corruption during the O2O phase. We hope this work inspires future research addressing data corruption in more realistic scenarios. A limitation of our approach is that IPW in RPEX requires access to the exact action probabilities, which may hinder integration with probability-agnostic models such as diffusion models.

## Acknowledgments and Disclosure of Funding

This work was supported by the Natural Science Foundation of Shenzhen (No.JCYJ20230807111604008, No.JCYJ20240813112007010), the Natural Science Foundation of Guangdong Province (No.2024A1515010003), National Key Research and Development Program of China (No.2022YFB4701400), and Cross-disciplinary Fund for Research and Innovation (No. JC2024002) of Tsinghua SIGS. Li Shen is supported by NSFC Grant (No. 62576364), Shenzhen Basic Research Project (Natural Science Foundation) Basic Research Key Project (NO. JCYJ20241202124430041), CCF-Tencent Rhino-Bird Open Research Fund (NO. CCF-Tencent RAGR20250114) and Tencent JR2025TEG002.

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

**Border Impact**. Offline-to-Online reinforcement learning (O2O RL) aims to learn a policy from a fixed offline dataset, followed by fine-tuning through online interaction. Our approach focuses on enhancing the robustness of O2O algorithms under data corruption occurring in both offline and online phases. Such corruption is ubiquitous due to noise and potential adversarial attacks. By improving robustness, our method advances the field of O2O RL and facilitates its deployment in real-world applications such as robotic control, without introducing significant ethical or societal concerns.

## Index of the Appendix

In the following, we briefly recap the contents of the Appendix.

– Appendix A presents further discussion of related work and additional background information.

– Appendix B reports all proofs and derivations.

– Appendix C reports additional experiments and ablation studies about our methods, along with some implementation details.

– Appendix D reports further discussion on the connection between the heavy-tailedness of the Q-target distribution and that of the learned policy.

## A    Related Works

**IPW in Heavy-tailedness Classification tasks**. Inverse probability weighting (IPW) [42] is a general methodology in survey sampling and causal inference [23], and is also widely employed to mitigate heavy-tailedness in classification tasks [7, 52]. In addressing heavy-tailed classification problems, IPW is typically applied as a weighting mechanism for the classification loss. For example, Cui et al. [7], Dang et al. [8] reweight the loss of a sample inversely proportional to the number of instances in its class. In this study, we incorporate IPW into the O2O reinforcement learning framework to address the heavy-tailed nature of policies under various data corruptions, and we demonstrate its significant effectiveness.

**Offline RL**. Offline RL algorithms need to avoid OOD actions. Previous methods to mitigate this issue under the model-free offline RL setting generally fall into three categories: 1) value function-based approaches, which implement pessimistic value estimation by assigning low values to out-of-distribution actions [27, 12], or implicit TD backups [26, 32] to avoid the use of out-of-distribution actions 2) sequential modeling approaches, which casts offline RL as a sequence generation task with return guidance [6, 22, 29, 1], and 3) constrained policy search (CPS) approaches, which regularizes the discrepancy between the learned policy and behavior policy [41, 40, 35, 19, 18].

**Kurtosis Value**. For a random variable $X$ with mean $\mu$ and std $\sigma$, metric Kurtosis [13, 33] is calculated by $\mathrm{Kurt}[X] = \mathbb{E}\left[\left(\frac{X-\mu}{\sigma}\right)^4\right]$, which measures the heavy-tailedness compared to a normal distribution. (Small Kurtosis means less heavy-tailedness) The excess kurtosis is defined as kurtosis minus 3. High values of $\mathrm{Kurt}$ arise in two circumstances: (1) where the probability mass is concentrated around the mean and the data-generating process produces occasional values far from the mean, (2) where the probability mass is concentrated in the tails of the distribution. (Note that the kurtosis reported in this paper is computed using SciPy [46], which, in fact, calculates the excess kurtosis.)

**AlignIQL** AlignIQL [17] aims to find the implicit policy in the learned value function of IQL [25]. To find the implicit policy, He et al. [17] define such implicit policy finding problem (IPF)

$$
\begin{aligned}
\min_{\pi} \quad & \mathbb{E}_{\boldsymbol{s}\sim d^{\pi}(\boldsymbol{s}), \boldsymbol{a}\sim\pi(\boldsymbol{a}|\boldsymbol{s})}\left[f\left(\frac{\pi(\boldsymbol{a}|\boldsymbol{s})}{\mu(\boldsymbol{a}|\boldsymbol{s})}\right)\right] \\
s.t. \quad & \pi(\boldsymbol{a}|\boldsymbol{s}) \geq 0, \quad \forall \boldsymbol{s}, \forall \boldsymbol{a} \\
& \int_{\boldsymbol{a}} \pi(\boldsymbol{a}|\boldsymbol{s})d\boldsymbol{a} = 1, \quad \forall \boldsymbol{s} \\
& \mathbb{E}_{\boldsymbol{a}\sim\pi(\boldsymbol{a}|\boldsymbol{s})}\left[Q(\boldsymbol{s}, \boldsymbol{a})\right] - V(\boldsymbol{s}) = 0, \quad \forall \boldsymbol{s},
\end{aligned}
\tag{IPF}
$$

where $V(s), Q(s, a)$ is the learned value function, which does not have to be the optimal value function. $f(\cdot)$ is a regularization function which aims to avoid out-of-distribution actions. Further He et al. [17] consider the relax formulation of Eq. (IPF)

$$\min_{\pi, V(s)} \mathbb{E}_{\substack{s \sim d^\pi(s) \\ a \sim \pi(a|s)}} \left[ f\left(\frac{\pi(a|s)}{\mu(a|s)}\right) + \eta \left(Q(s, a) - V(s)\right)^2 \right]$$

$$s.t. \quad \pi(a|s) \geq 0, \quad \forall s, \forall a \qquad \text{(IPF-Soft)}$$

$$\int_a \pi(a|s) da = 1, \quad \forall s,$$

and derive the formulation of implicit policy (suppose that $f(x) = \log x$)

$$\pi^\star(a|s) \propto \mu(a|s) \exp\left\{ -\eta \left(Q(s, a) - V(s)\right)^2 \right\}. \qquad (10)$$

# B Proofs and Derivations

In this section, we present a detailed derivation and justification of our method, along with the proofs of the main theoretical results.

## B.1 Derivation and Theoretical Justification of RPEX

Following Zhang et al. [53], let $K = 2$, the compositional policy $\hat{\pi}$ can be rewritten as

$$\tilde{\pi}(a|s) = \int_{a_1, a_2} \pi_\beta(a_1|s) \pi_\theta(a_2|s) \sum_i P_{\mathbf{w}}(i|s, a_1, a_2) \delta(a = a_i) da_1 da_2, \qquad (11)$$

where $\delta$ is the Dirac delta function, $P_{\mathbf{w}}$ is a discrete policy with action dimension of two. For V-value, we can have the following derivations (Zhang et al. [53], Appendix A.3)

$$V(s) = \int_a \tilde{\pi}(a|s) Q_\phi(s, a) da \qquad (12)$$

$$= \int_a \left[ \int_{a_1, a_2} \pi_\beta(a_1|s) \pi_\theta(a_2|s) \sum_i P_{\mathbf{w}}(i|s, a_1, a_2) \delta(a = a_i) da_1 da_2 \right] Q_\phi(s, a) da \qquad (13)$$

$$= \int_{a_1, a_2} \pi_\beta(a_1|s) \pi_\theta(a_2|s) \sum_i P_{\mathbf{w}}(i|s, a_1, a_2) \left[ \int_a \delta(a = a_i) Q_\phi(s, a) da \right] da_1 da_2 \qquad (14)$$

$$= \int_{a_1, a_2} \pi_\beta(a_1|s) \pi_\theta(a_2|s) \sum_i P_{\mathbf{w}}(i|s, a_1, a_2) Q(s, a_i) da_1 da_2 \qquad (15)$$

$$= \mathbb{E}_{a_1 \sim \pi_\beta, a_2 \sim \pi_\theta} \left[ \sum_i P_{\mathbf{w}}(i|s, a_1, a_2) Q_\phi(s, a_i) \right], \qquad (16)$$

where $a_1 \sim \pi_\beta, a_2 \sim \pi_\theta$.

By incorporating an entropy regularization term to promote online exploration, we obtain

$$\mathbb{E}_{a_1 \sim \pi_\beta, a_2 \sim \pi_\theta} \left[ \sum_i P_{\mathbf{w}}(i|s, a_1, a_2) \left(Q_\phi(s, a_i) - \alpha \log P_{\mathbf{w}}(i|s, a_1, a_2)\right) \right]. \qquad (17)$$

By adding a regularization term

$$\kappa_1 \frac{Q_\phi - V}{\pi_i(a_i|s)} P_{\mathbf{w}}(i|s, a_1, a_2), \qquad (18)$$

where $\kappa_1$ is the coefficient. $\pi_i(a_i|s)$ is the probability that action $a_i$ is selected by the corresponding policy $\pi_i$. $\pi_1$ denotes the offline policy $\pi_\beta$ and $\pi_2$ denotes the online policy $\pi_\theta$.

We obtain

$$\mathbb{E}_{\substack{a_1 \sim \pi_\beta, \\ a_2 \sim \pi_\theta}} \left[ \sum_i P_{\mathbf{w}}(i|s, a_1, a_2) \left( Q_\phi(s, a_i) - \alpha \log P_{\mathbf{w}}(i|s, a_1, a_2) \right) + \kappa_1 \frac{Q_\phi - V}{\pi_i(a_i|s)} P_{\mathbf{w}}(i|s, a_1, a_2) \right]. \tag{19}$$

Solving $P_{\mathbf{w}}$ by setting the gradient of Eq. (19) to 0 with respect to $P_{\mathbf{w}}$ given $\pi_\beta, \pi_\theta$, we have

$$P_{\mathbf{w}}(i|s, a_1, a_2) \propto \exp \left( Q_\phi(s, a_i)/\alpha + \kappa_1/\alpha \frac{Q_\phi - V}{\pi_i(a_i|s)} \right). \tag{20}$$

Finally, let $\kappa = \kappa_1/\alpha$, we have (for simplicity, we omit the subscripts $\phi$ and $\psi$.)

$$P_{\mathbf{w}}[i] = \frac{\exp(Q(s, a_i)/\alpha) + \kappa \frac{Q-V}{\pi_i(a_i|s)}}{\sum_j \exp(Q(s, a_j)/\alpha) + \kappa \frac{Q-V}{\pi_j(a_j|s)}}, \quad \forall i \in [1, \cdots, K], \tag{21}$$

which is the Eq. (6).

## C   Implementation Details and Additional Experiments

In this section, we introduce the implementation details for reproducing our results and some extra experiments to validate our method.

### C.1   Implementation Details

**Evaluation**. Throughout this paper, we report mean scores and standard deviation averaged over the last three evaluations across different random seeds. Each evaluation consists of 10 episodes.

To obtain the offline checkpoints for RIQL and IQL, we rerun the official code released by Yang et al. [50] with default configuration parameters. The offline policy and value function are trained for 2e6 gradient steps, while the O2O update phase consists of 1e6 steps. For O2O results, we use the official code from Zhang et al. [53] to implement our RIQL+PREX and IQL+RPEX methods. We adopt the codebases of Zhang et al. [53] and Yang et al. [50] to implement the baseline methods. For RIQL-based O2O implementations, following Yang et al. [50], we employ the Huber loss and Q quantile estimators. The policy and value networks for both IQL and RIQL are implemented as multilayer perceptrons (MLPs) with two hidden layers, each comprising 256 units and ReLU activations. For Q-network in RIQL-based methods, we follow RIQL and adopt the ensemble implementation of EDAC [2]. For RPEX with a deterministic offline policy, we parameterize the online policy within the policy set $\Pi$ using a stochastic Gaussian policy and assign the IPW weight of the online policy to the offline deterministic policy. These neural networks are updated using the Adam [24]. The hyperparameters used to reproduce our results in Table 1 are listed in Table C.1.

The following section provides details on the baseline implementations.

**State Normalization**. Following Yang et al. [50], we normalize each state in the corrupted offline dataset and environment using $\mu_o$ and $\sigma_o$, which denote the mean and standard deviation of all states and next states in the corrupted offline replay buffer $\hat{\mathcal{D}}_{\text{offline}}$. Based on $\mu_o$ and $\sigma_o$, the states and next-states are normalized as $s_i = \frac{s_i - \mu_o}{\sigma_o}$, $s_i' = \frac{s_i' - \mu_o}{\sigma_o}$, where $\mu_o = \frac{1}{2N} \sum_{i=1}^{N}(s_i + s_i')$ and $\sigma_o^2 = \frac{1}{2N} \sum_{i=1}^{N} \left[ (s_i - \mu_o)^2 + (s_i' - \mu_o)^2 \right]$.

**RIQL-direct**. For RIQL, we directly use the offline pretrained policy and critic for online learning. Accordingly, the offline policy and critics continue to be updated during the online phase with additional samples collected by the offline policy. We adopt the offline parameters reported by Yang et al. [50] and apply the same settings during online training. For the results shown in Table 1, we sweep over UTD values $\{1, 2, 4\}$ and report the best outcome.

**RIQL-PEX**. For RIQL-PEX, the critic and policy updates follow the same procedure as RIQL, except that online samples are collected using a composited policy (Eq. (5)). Following PEX [53], we freeze the offline policy and train a new policy using both offline and online buffers. For evaluation, consistent with Zhang et al. [53], we greedily select the action with the highest weight according to

Eq. (5). For the results in Table 1, we sweep over UTD values $\{1, 2, 4\}$ and $\alpha \in \{3, 10, 100\}$, and report the best performance.

**IQL**. For IQL, we also directly use the offline pretrained policy and critic for online learning. The offline policy and critics are further updated during the online phase with samples collected by the offline policy. As with RIQL, we sweep over UTD values $\{1, 2, 4\}$ and report the best result presented in Table 1.

**IQL-PEX**. For IQL-PEX, the critic and policy updates follow the same scheme as RIQL, with online samples collected using the composited policy (Eq. (5)). Following PEX [53], we freeze the offline policy and train a new policy with both offline and online data. For evaluation, we greedily select the highest-weighted action according to Eq. (5), consistent with Zhang et al. [53]. As with RIQL-PEX, we sweep over UTD values $\{1, 2, 4\}$ and $\alpha \in \{3, 10, 100\}$, and report the best performance.

**Data Corruption Details**. Following Yang et al. [50], we apply both random and adversarial corruption to the four components: states, actions, rewards, and dynamics (i.e., next states). The overall corruption level is governed by four parameters: $c_1$, $c_2$, $\epsilon_1$, and $\epsilon_2$, where $c_1$ denotes the corruption rate within an offline dataset of size $N$, $c_2$ represents the corruption probability of online transitions, $\epsilon_1$ specifies the offline corruption scale per dimension, and $\epsilon_2$ specifies the online corruption scale per dimension. Our experiments primarily focus on the "medium-replay" datasets introduced by Fu et al. [10], which are collected during the training of a Soft Actor-Critic (SAC) agent and are thus more representative of real-world scenarios. Unless otherwise specified, we set $c_1 = 0.3$, $c_2 = 0.5$, and $\epsilon_1 = \epsilon_2 = 1$. Below, we describe four types of random data corruption, as well as a mixed corruption strategy.

- **Random observation attack**: For offline attacks, we randomly sample $c_1 \cdot N$ transitions $(\boldsymbol{s}, \boldsymbol{a}, r, \boldsymbol{s}')$ and corrupt the states as $\hat{\boldsymbol{s}} = \boldsymbol{s} + \lambda \cdot \text{std}(\boldsymbol{s})$, where $\lambda \sim \text{Uniform}[-\epsilon, \epsilon]^{d_{\boldsymbol{s}}}$. Here, $d_{\boldsymbol{s}}$ denotes the dimensionality of the state, and $\text{std}(\boldsymbol{s})$ is the $d_{\boldsymbol{s}}$-dimensional standard deviation of all states in the offline dataset. The noise is independently added to each dimension and scaled by the corresponding standard deviation. For online observation attacks, we randomly corrupt online $\boldsymbol{s}$ with probability $c_2$. If state normalization is enabled, we set $\text{std}(\boldsymbol{s}) = 1$; otherwise, we use the standard deviation computed from the offline dataset.

- **Random action attack**: For offline attacks, we randomly select $c_1 \cdot N$ transitions $(\boldsymbol{s}, \boldsymbol{a}, r, \boldsymbol{s}')$ and corrupt the action as $\hat{\boldsymbol{a}} = \boldsymbol{a} + \lambda \cdot \text{std}(\boldsymbol{a})$, where $\lambda \sim \text{Uniform}[-\epsilon, \epsilon]^{d_{\boldsymbol{a}}}$. Here, $d_{\boldsymbol{a}}$ denotes the dimensionality of the action, and $\text{std}(\boldsymbol{a})$ is the $d_{\boldsymbol{a}}$-dimensional standard deviation of all actions in the offline dataset. For online action attacks, we randomly corrupt $\boldsymbol{a}$ with probability $c_2$, using $\text{std}(\boldsymbol{a})$ computed from the offline dataset.

- **Random reward attack**: For offline attacks, we randomly sample $c_1 \cdot N$ transitions $(\boldsymbol{s}, \boldsymbol{a}, r, \boldsymbol{s}')$ from $D$ and corrupt the reward as $\hat{r} \sim \text{Uniform}[-30 \cdot \epsilon, 30 \cdot \epsilon]$. For online reward attacks, we randomly corrupt $r$ with probability $c_2$ using the same corruption method as in the offline case.

- **Random dynamics attack**: For offline attacks, we randomly sample $c_1 \cdot N$ transitions $(\boldsymbol{s}, \boldsymbol{a}, r, \boldsymbol{s}')$ and corrupt the next state as $\hat{\boldsymbol{s}}' = \boldsymbol{s}' + \lambda \cdot \text{std}(\boldsymbol{s}')$, where $\lambda \sim \text{Uniform}[-\epsilon, \epsilon]^{d_{\boldsymbol{s}}}$. Here, $d_{\boldsymbol{s}}$ denotes the dimensionality of the state, and $\text{std}(\boldsymbol{s}')$ is the $d_{\boldsymbol{s}}$-dimensional standard deviation of all next states in the offline dataset. For online observation attacks, we randomly corrupt the online next state $\boldsymbol{s}'$ with probability $c_2$. If state normalization is enabled, we set $\text{std}(\boldsymbol{s}') = 1$; otherwise, we use the standard deviation computed from the offline dataset.

- **Random mixed attack**: The offline dataset is corrupted using random dynamics attacks. During the online phase, we randomly corrupt $r$ and $\boldsymbol{s}'$ with probability $c_2$, applying the same corruption method as described previously.

In addition, we apply online adversarial dynamics attacks:

- **Adversarial dynamics attack**: Following Yang et al. [50], we use a pretrained EDAC [2] agent consisting of a set of $Q_p$ functions and a policy function $\pi_p$. We then randomly corrupt $\boldsymbol{s}'$ with probability $c_2$ and modify the next state as $\hat{\boldsymbol{s}}' = \arg\min_{\hat{\boldsymbol{s}}' \in \mathbb{B}_d(\boldsymbol{s}', \epsilon)} Q_p(\hat{\boldsymbol{s}}', \pi_p(\hat{\boldsymbol{s}}'))$, where $\mathbb{B}_d(\boldsymbol{s}', \epsilon) = \{\hat{\boldsymbol{s}}' \mid |\hat{\boldsymbol{s}}' - \boldsymbol{s}'| \leq \epsilon \cdot \text{std}(\boldsymbol{s}')\}$. The $Q$ function in the objective is defined as the average over the $Q$ functions in EDAC. We implement the optimization using Projected Gradient Descent, following prior work [50, 54]. Specifically, we initialize a learnable vector $z \in [-\epsilon, \epsilon]^{d_{\boldsymbol{s}}}$ and perform two steps of gradient descent with a step size of 0.1 on $\hat{\boldsymbol{s}}' = \boldsymbol{s}' + z \cdot \text{std}(\boldsymbol{s}')$. After each update, we clip each dimension of $z$ to the range $[-\epsilon, \epsilon]$.

**Toy Experiments Details**. For the toy maze experiment, we train IQL and IQL+IPW using the offline dataset shown in Figure 2(a). The reward is defined as the negative Euclidean distance between the current position and the goal. The action space consists of eight movement directions, and the state is represented by the current coordinates of the agent. In the corrupted setting, random noise is added to the dataset states with a corruption range of 1 and a corruption rate of 0.5.

| | |
|---|---|
| **LR** (For all networks) | 3e-4 |
| **Critic Batch Size** | 256 |
| **Actor Batch Size** | 256 |
| **Discount** | 0.99 |
| **Offline Batch Ratio** | 0.5 |
| $\tau$ **Expectiles for IQL or RIQL** | 0.7 (MuJoCo) |
| **Initial Collection Steps** | 5000 |
| **Target Update Speed** | 5e-3 |
| **IPW Weight** $\kappa$ | 0.1 (except mixed attack) |
| **IPW Weight** $\kappa$ | 0.3 (mixed attack) |
| **Inverse Temperature** $\alpha^{-1}$ | 3 (except mixed attack) |
| **Inverse Temperature** $\alpha^{-1}$ | 10 (mixed attack) |
| **UTD for RPEX** | 4 for action corruption 1 for others |
| **Policy Extraction for RPEX** | Eq. (8) for state attack |
| **Policy Extraction for RPEX** | Eq. (3) (except state attack) |
| $\eta$ **for AlignIQL** | 3 |
| **State Normalization** | True |
| **Number of Offline Iterations** | 2e6 |
| **Number of Online Iterations** | 1e6 |
| **Optimizer** | Adam [24] |

## C.2 Ablation Study

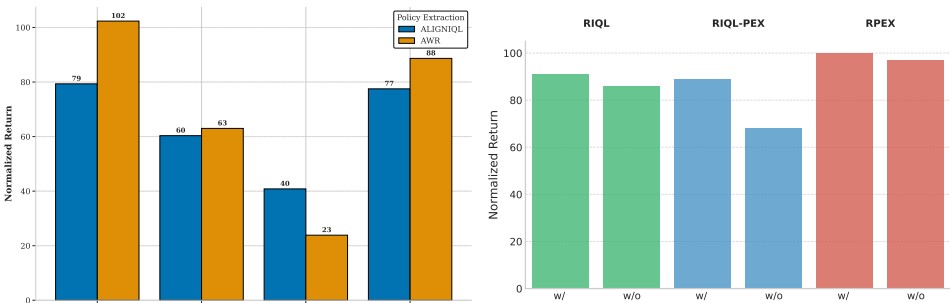

Figure 6: Policy Extraction Ablation.

Figure 7: Offline buffer Ablation.

**High UTD on Critic** We conduct experiments in which only the critic is trained with a high update-to-data (UTD) ratio, while the actor maintains UTD = 1, on the Walker2d-MR tasks under dynamics and action corruption. All other hyperparameters remain the same as those used in our main paper (Table C.1). Table 2 shows the results of RPEX with such UTD implementation. All results are averaged over 10 random seeds.

Table 2: Results under High UTD for the Critic.

| UTD | Dynamics | Action |
|---|---|---|
| 1 | $80 \rightarrow 92.2$ ±1.4 | $85.9 \rightarrow 118.9$ ±10.1 |
| 4 | $80 \rightarrow 92.6$ ±10.0 | $85.9 \rightarrow 96.7$ ±21.8 |

As shown in Table 2, the issue discussed in Section 7.2—namely, that high UTD primarily improves robustness against action corruption while slightly degrading overall performance—can be alleviated. However, we observe that although such UTD implementations can sometimes yield better performance, they also increase overall variance. This phenomenon may be attributed to the fact that data corruption introduces greater Bellman backup errors compared to the non-corrupted setting. While a high UTD ratio for the critic can accelerate learning, the subsequent Bellman backups may fluctuate significantly due to corrupted data.

**Policy Extraction Methods** In this section, we investigate the effect of policy extraction in our method. Since IQL-style methods follow an actor-critic decoupled paradigm, different policy extraction methods can significantly affect final performance [17, 39, 16].

As shown in Fig.6, extracting the policy using AlignIQL[17] significantly improves robustness under state observation corruption, whereas in other scenarios, the vanilla AWR method yields better performance.

**Buffer** In this part, we investigate the impact of retaining offline datasets during online fine-tuning. As noted in Zhou et al. [56], most reinforcement learning fine-tuning methods rely on continued access to offline data to ensure stability and performance. As shown in Fig. 7, the presence or absence of offline data during the fine-tuning phase has minimal impact on our method, demonstrating its robustness.

**Range of Clip** For range of clip, we conduct ablation study on the Walker2d-MR environment under dynamics and action corruption. The reason to choose dynamics and action corruption is that they represent two modular $s, a$ in the MDP. As shown in RIQL, dynamics corruption is known as difficulty for robust RL. The results of different clip ranges on the Walker2d-MR are shown in the Table 3. Note that the $\kappa = 0.1$, $\alpha^{-1} = 3$, corruption range (1) and rate (0.5) are the same as those in Table 1. All results are averaged over 5 random seeds.

Table 3: Ablation Study of the Clip Range.

| Clip Range | Dynamics | Action |
|---|---|---|
| (-10000, 100) | $80 \rightarrow 92.2$ ±1.4 | $85.9 \rightarrow 118.9$ ±10.1 |
| (-100, 100) | $80 \rightarrow 90.8$ ±30.0 | $85.9 \rightarrow 108.4$ ±1.07 |
| (0, 100) | $80 \rightarrow 62.1$ ±38.6 | $85.9 \rightarrow 62.67$ ±44.2 |
| (-10000, 10000) | $80 \rightarrow 96.7$ ±1.69 | $85.9 \rightarrow 105.53$ ±0.09 |

As shown in Table 3, the performance differences across various clipping ranges are minor, except for extreme cases such as the range (0, 100). Similar to $\kappa$, fine-tuning the clipping range may further improve performance. Throughout our paper, we consistently use the same values for $\kappa$ and the clipping range $(-10000, 100)$, which further demonstrates the robustness of RPEX.

### C.3  Additional Experiments

**Training Time** We report the average epoch time for the Hopper-MR task as a measure of computational cost in Table 1. All experiments were conducted on an NVIDIA RTX 4090 24GB GPU with an Intel(R) Xeon(R) Platinum 8358P CPU. As shown in Table 4, RIQL [50] incurs additional computational overhead compared to IQL due to the Huber loss and quantile Q estimators. PEX [53] also requires more computation than IQL as it involves training an additional online policy. RPEX introduces further overhead relative to PEX owing to the added computation of IPW. Nevertheless, despite achieving a performance improvement of approximately 13.6% over PEX, RPEX increases

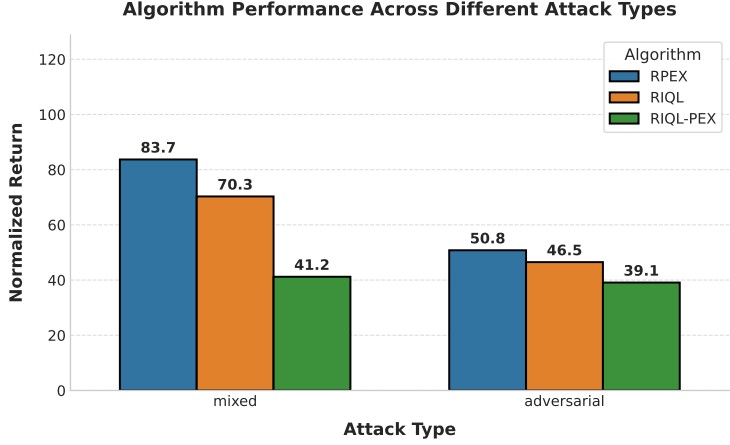

Figure 8: Results under random mixed and adversarial attacks averaged over 3 random seeds.

training time by only about 8.1%. These results indicate that our method improves performance without incurring substantial computational costs.

Table 4: Average Training Cost.

| IQL | IQL-PEX | IQL-RPEX (ours) | RIQL | RIQL-PEX | RIQL-RPEX (ours) |
|---|---|---|---|---|---|
| $17.4_{\pm0.05}$ | $30.9_{\pm3.2}$ | $31.7_{\pm1.6}$ | $22.7_{\pm1.3}$ | $33.1_{\pm1.9}$ | $37.6_{\pm1.3}$ |

**Adversarial Corruption & Mixed Corruption** Figure 8 presents the performance of RIQL, RIQL-PEX, and RPEX (ours) under random mixed and adversarial attacks in Hopper-MR tasks. For random mixed attacks, the offline policy is trained with a random dynamics attack, followed by online random dynamics and reward attacks. For adversarial attacks, the offline policy is trained with a random dynamics attack, followed by online adversarial dynamics attacks. For RIQL, we sweep from UTD$\in \{1, 4\}$ and report the best results. For RIQL-PEX, we perform a hyperparameter sweep over UTD $\in \{1, 4\}$ and inverse temperature $\alpha^{-1} \in \{3, 10, 100\}$, and report the best results. For our method, we sweep over $\kappa \in \{0.1, 0.3\}$ and $\alpha^{-1} \in \{3, 10\}$.

As shown in Figure 8, RPEX outperforms other methods under both mixed and adversarial attack settings.

**AntMaze Tasks** We conducted additional experiments on the AntMaze-large and AntMaze-diverse tasks, which are the most challenging tasks in the AntMaze tasks. We primarily focus on dynamics and observation corruption, as these represent the most difficult corruption settings.

**Why are observation and dynamics corruptions particularly challenging?** As shown in Table 1, all algorithms struggle to improve under observation and dynamics attacks. In particular, under observation attack, RIQL-PEX exhibits a performance drop of approximately 41% on the Hopper-MR task. Although RPEX significantly outperforms RIQL-PEX, it still struggles to improve performance. A similar phenomenon is also reported in RIQL [50].

The primary cause of poor performance under observation attack is its strong disruption to online exploration. The key objective during the online phase is to collect optimal trajectories and update both the policy and the value function. When observation attacks occur, they affect action selection, potentially resulting in poor or even out-of-distribution (OOD) actions. Moreover, observation attacks can distort the Bellman backup, thereby hindering value function learning. According to Theorem 3 in RIQL, such suboptimal or OOD actions, combined with high Bellman backup error, lead to a loose upper bound on the difference between value functions learned under clean and corrupted observations. This explains why improving performance under observation attacks is particularly difficult.

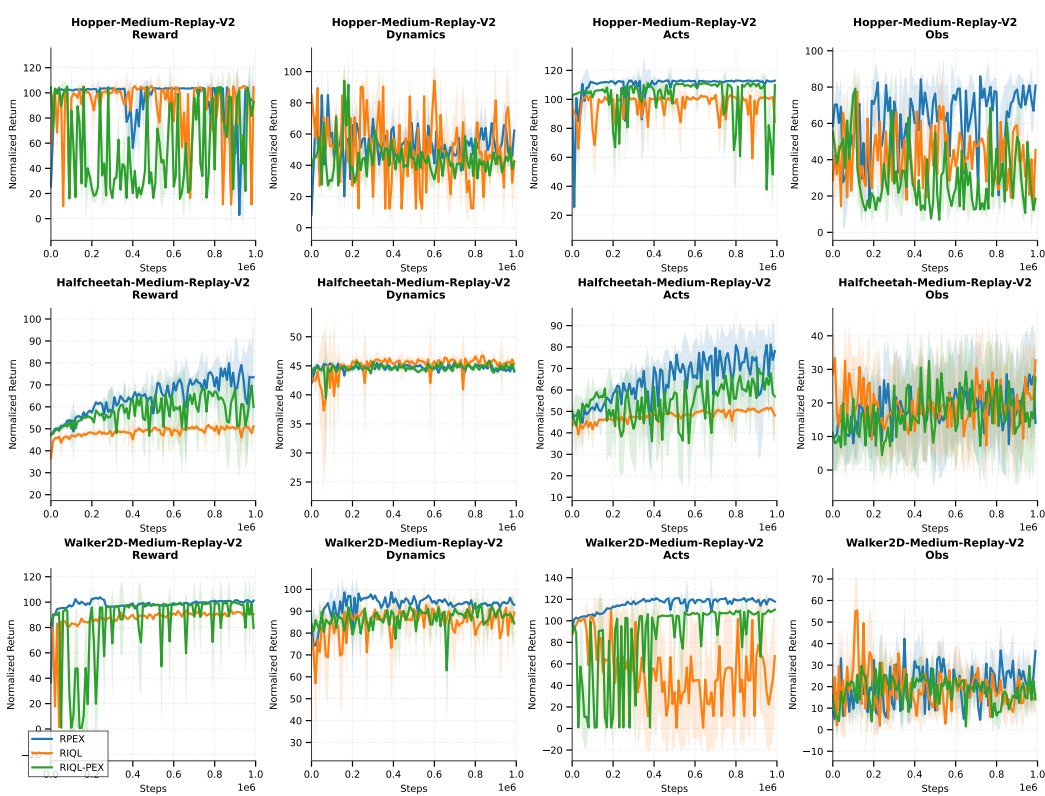

Figure 9: Training curves of RIQL-direct, RIQL-PEX (RIQL), and RPEX (ours) under random data corruption on the "medium-replay" tasks (Table 1).

For dynamics attacks, the limited improvement after the O2O phase primarily stems from Bellman backup errors induced by the attack. Although RIQL reduces Bellman backup error by mitigating the heavy-tailedness of the Q-target, attacks on the dynamics during the online phase still distort the Bellman backup, thereby limiting performance improvement.

Given the difficulty of observation and dynamics corruptions, together with the sparse-reward challenge in AntMaze tasks, we adopt a corruption range of 0.3 and a corruption rate of 0.2 during offline pretraining, consistent with RIQL. For online corruption, we apply a corruption range of 0.5 and a corruption rate of 0.3. The offline value function is trained using RIQL with the default hyperparameters reported in the original RIQL paper. In the AntMaze experiments, we employ the same hyperparameters as those in the main paper (Table C.1) to demonstrate the robustness of RPEX. All results are averaged over 10 random seeds.

Table 5: Results on AntMaze Large Tasks under Dynamics and Observation Corruption.

| Environment | RIQL-PEX | RPEX |
|---|---|---|
| Large-play Dynamics | $33.3 \rightarrow 37.8$ ±4.2 | $\mathbf{33.3 \rightarrow 38.9}$ ±1.8 |
| Large-play Observation | $23.3 \rightarrow 21.1$ ±7.1 | $\mathbf{23.3 \rightarrow 35.3}$ ±2.7 |
| Large-diverse Dynamics | $23.3 \rightarrow 20.0$ ±5.4 | $\mathbf{23.3 \rightarrow 25.6}$ ±4.7 |
| Large-diverse Observation | $26.7 \rightarrow 25.6$ ±8.1 | $\mathbf{26.7 \rightarrow 36.7}$ ±5.3 |
| Average | $26.6 \rightarrow 26.1$ | $\mathbf{26.6 \rightarrow 34.1}$ |

As shown in Table 5, RPEX outperforms RIQL-PEX by a large margin of approximately 30.7%.

**Training Curves** The complete training curves for RIQL-direct, RIQL-PEX (RIQL), and RPEX are presented in Fig. 9.

**Varying Levels of Data Corruption**  We conduct experiments with varying online corruption levels (including the non-corrupted setting) on the Walker2d-MR tasks under both dynamics and action corruption. The results are presented in Table 6. We use the same hyperparameters as in the main paper, i.e., $\kappa = 0.1$, $\alpha^{-1} = 3$, etc. All results are averaged over 5 random seeds. Across varying corruption levels, RPEX demonstrates strong robustness compared to RIQL-PEX, outperforming it by an average of approximately 16.8%, further validating the effectiveness of RPEX.

Table 6: Results under varying levels of online corruption.

| $(\epsilon_2, c_2)$ | Dynamics (RIQL-PEX) | Action (RIQL-PEX) | Dynamics (RPEX) | Action (RPEX) |
|---|---|---|---|---|
| (0,0) | $80 \rightarrow 94.8$ ±13.7 | $85.9 \rightarrow 101.8$ ±0.2 | $\mathbf{80 \rightarrow 98.9}$ ±6.9 | $\mathbf{85.9 \rightarrow 102.5}$ ±4.6 |
| (2,0.3) | $80 \rightarrow 37.6$ ±12.5 | $85.9 \rightarrow 69.5$ ±29.3 | $\mathbf{80 \rightarrow 77.8}$ ±17.5 | $\mathbf{85.9 \rightarrow 97.0}$ ±105.3 |
| (1,0.5) | $80 \rightarrow 89.5$ ±1.3 | $85.9 \rightarrow 109.2$ ±15.6 | $\mathbf{80 \rightarrow 92.2}$ ±1.4 | $\mathbf{85.9 \rightarrow 118.9}$ ±10.1 |
| Average | $80 \rightarrow 74.0$ | $85.9 \rightarrow 93.5$ | $\mathbf{80 \rightarrow 89.6}$ | $\mathbf{85.9 \rightarrow 106.1}$ |

In the non-corrupted setting, $(\epsilon_2, c_2) = (0, 0)$, when the value function is inaccurate due to data corruption or limited coverage, some promising actions may be mistakenly assigned negative advantages, leading RPEX to penalize them. However, although the value functions in RPEX and other algorithms are trained on corrupted offline datasets, they nonetheless preserve substantial information that is beneficial for online exploration. If the offline value functions are made more robust, as in RIQL, the value estimates become more reliable.

Moreover, as illustrated in Figures 2(d)–(e), when the value functions are inaccurate, most penalized actions correspond to suboptimal actions generated by the heavy-tailed policy. While IPW may also penalize some promising actions, it improves overall exploration efficiency (Figure 2(e)). The above analysis demonstrates that the IPW constraint in RPEX does not substantially impair O2O exploration efficiency under uncorrupted settings. This viewpoint is further supported in Table 6, where RPEX outperforms RIQL-PEX in both the uncorrupted setting and across varying corruption rates. Moreover, such penalties can be tuned via $\kappa$ or the clipping range. In our experiments, we fix both $\kappa$ and the clipping range to demonstrate the inherent robustness of RPEX; fine-tuning these parameters could further improve exploration efficiency and performance.

# D   Further Discussion

In both RIQL and IQL, the policy is learned via Advantage Weighted Regression (AWR), which can be interpreted as a form of supervised learning weighted by the value function. In the non-corrupted setting, a well-estimated value function improves the policy by assigning higher weights to actions with positive advantages, without increasing the heavy-tailedness of the policy, as shown in Figure 1(b). However, as also illustrated in Figure 1(b), our experiments demonstrate that this conclusion does not hold under corrupted settings: heavy-tailed Q-targets increase the heavy-tailedness of the policy. Furthermore, merely mitigating the heavy-tailedness of the Q-function, as in RIQL, does not reduce the policy's heavy-tailedness.

Such heavy-tailed policies have limited impact during evaluation, as the policy is executed deterministically by selecting the action with the highest probability in the evaluation environment, which reduces the effect of policy heavy-tailedness. However, these heavy-tailed policies lead to inefficient online exploration, which in turn results in the performance degradation of RIQL after online fine-tuning (Table 1).

In summary, data corruption induces heavy-tailed Q-targets, and this heavy-tailedness is inherited by the policy through AWR, a supervised-like training paradigm. While the heavy-tailedness has limited influence during evaluation, it significantly impairs online exploration.

