# OpenReview forum: "Robust Policy Expansion for Offline-to-Online RL under Diverse Data Corruption"
_NeurIPS.cc/2025/Conference — NeurIPS 2025 poster_

### Official Review · Reviewer_REEm · 2025-06-29

**Clarity:** 2
**Significance:** 3
**Originality:** 2
**Rating:** 4
**Confidence:** 3

**Summary:**

This paper focuses on offline-to-online (O2O) reinforcement learning under various types of data corruption. The authors observe that such corruption can induce heavy-tailed behavior in the learned policy. To address this, they propose RPEX, a method that incorporates inverse probability weighting to adjust the action probabilities within the PEX framework. Experiments on D4RL Mujoco tasks demonstrate the effectiveness of the proposed approach in corrupted data settings.

**Questions:**

1. According to Eq. (6), when an action has a negative advantage and a small probability, the adjusted probability $P_w[i]$ could be negative.  In practice, is this value clipped to zero during implementation?

2. In lines 196–202, the text refers to Eq. (5), but based on the context, should this reference instead be to Eq. (6)?

**Ethical Concerns:**

["NO or VERY MINOR ethics concerns only"]

**Final Justification:**

Overall, I am inclined to support this paper. While the proposed method is incremental, it demonstrates clear effectiveness under the diverse data corruption setting, and also shows benefits on non-corrupted data compared with PEX.

**Limitations:**

see Weaknesses

**Quality:**

3

**Strengths And Weaknesses:**

Strengths:

1.Investigating O2O RL under data corruption is an important and practically relevant direction, as such scenarios are common in real-world applications.

2.The proposed method is simple yet effective, and its design based on the PEX framework allows for integration with various offline RL algorithms.

3.The paper is clearly written, and the experiments effectively support the existence of the heavy-tailedness issue in the policy trained on corrupted data.

Weaknesses:

1.The paper lacks a thorough discussion on the connection between the heavy-tailedness of the Q-target distribution and that of the learned policy. One confusing aspect is that RIQL, which explicitly addresses heavy-tailed Q-targets, still exhibits higher policy heavy-tailedness as shown in Figure 1(b). Moreover, RIQL—despite having greater policy heavy-tailedness—outperforms IQL in the experiments. This appears to contradict the paper's conclusion that heavy-tailed policies hinder efficient exploration.

2.The paper does not include experiments under non-corrupted data settings. The proposed method constrains the probability of actions with negative advantage estimates; however, when value function estimation is inaccurate, such constraints might prevent exploration of promising actions, potentially leading to slower performance improvement.

3.The experimental section lacks an analysis of performance under varying levels of data corruption. Evaluating the method's robustness across different corruption rates would provide stronger empirical support and clarify the boundary of its effectiveness.

---

> ### Author Rebuttal · Authors · 2025-07-28
>
> We are grateful to the reviewer for their time and effort in reviewing our paper, as well as for the constructive suggestions. Below, we address your concerns one by one.
>
> > Why RIQL despite having greater policy heavy-tailedness—outperforms IQL in the experiments.
>
> Thank you for pointing this out. This is a very insightful question. RIQL addresses heavy-tailed Q-targets through observation normalization, the Huber loss, and quantile Q estimators. However, as shown in Figure 1(b), RIQL increases the heavy-tailedness of the policy, and such heavy-tailed policies can lead to inefficient exploration (Figure 2(d)). The reason RIQL exhibits greater policy heavy-tailedness yet outperforms IQL is that “the policy is evaluated deterministically by selecting the action with the highest probability in the evaluation environments, which limits the influence of policy heavy-tailedness” (lines 167–169). RIQL learns a better Q-function than IQL in corrupted settings, and the impact of heavy-tailedness is mitigated during evaluation for the reason stated above. This interpretation is supported by the O2O results in Table 1. In Table 1, vanilla RIQL, when trained with additional data during the online phase, experiences a performance degradation of approximately 6.1%—the only algorithm to show a decline in performance following online fine-tuning—whereas IQL improves by about 13.1%. The performance degradation of RIQL during the O2O phase is attributed to poor online exploration caused by policy heavy-tailedness. Therefore, this phenomenon does not contradict our conclusion that heavy-tailed policies hinder efficient exploration.
>
> > lack of discussion on the connection between the heavy-tailedness of the Q-target distribution and that of the learned policy.
>
> Thank you for pointing this out. This is a very insightful question. In both RIQL and IQL, the policy is learned via Advantage Weighted Regression (AWR), which can be interpreted as a form of supervised learning weighted by the value function. In the non-corrupted setting, a well-estimated value function improves the policy by assigning higher weights to actions with positive advantages, without increasing the heavy-tailedness of the policy, as shown in Figure 1(b). However, as also illustrated in Figure 1(b), our experiments demonstrate that this conclusion does not hold under corrupted settings: heavy-tailed Q-targets increase the heavy-tailedness of the policy. Furthermore, merely mitigating the heavy-tailedness of the Q-function, as in RIQL, does not reduce the policy’s heavy-tailedness.
>
> Such heavy-tailed policies have limited impact during evaluation, as the policy is executed deterministically by selecting the action with the highest probability in the evaluation environment, which reduces the effect of policy heavy-tailedness (lines 167–169). However, these heavy-tailed policies lead to inefficient online exploration, which in turn results in the performance degradation of RIQL after online fine-tuning (Table 1).
>
> In summary, data corruption induces heavy-tailed Q-targets, and this heavy-tailedness is inherited by the policy through AWR, a supervised-like training paradigm. While the heavy-tailedness has limited influence during evaluation, it significantly impairs online exploration. We will incorporate this discussion into the revised version of our paper.
>
> > experiments under non-corrupted data setting and varying levels of data corruption
>
> Thank you for pointing this out. We conduct experiments with varying online  corruption levels (including the non-corrupted setting) on the Walker2d-MR tasks under both dynamics and action corruption. The results are presented in the following table. We use the same hyperparameters as in the main paper, i.e., $\kappa = 0.1$, $\alpha^{-1} = 3$, etc. All results are averaged over 5 random seeds.
>
> | Online corruption level: (range,rate) | dynamics (RIQL-PEX) | action (RIQL-PEX)     | dynamics (RPEX)        | action (RPEX)             |
> | ------------------------------------- | ------------------- | --------------------- | ---------------------- | ------------------------- |
> | (0,0)                                 | 80->94.8$\pm$ 13.7  | 85.9->101.8$\pm$ 0.2  | **80->98.9$\pm$ 6.9**  | **85.9->102.5$\pm$ 4.6**  |
> | (2,0.3)                               | 80->37.6$\pm$ 12.5  | 85.9->69.5$\pm$ 29.3  | **80->77.8$\pm$ 17.5** | **85.9->97.0$\pm$ 105.3** |
> | (1,0.5)                               | 80->89.5$\pm$ 1.3   | 85.9->109.2$\pm$ 15.6 | **80->92.2$\pm$ 1.4**  | **85.9->118.9$\pm$ 10.1** |
> | Average                               | 80->74.0            | 85.9->93.5            | **80->89.6**           | **85.9->106.1**           |
>
> As shown in the table above, RPEX outperforms RIQL-PEX in both the non-corrupted setting and across varying corruption rates.
>
> In the non-corrupted setting, when the value function is inaccurate due to data corruption or limited coverage, some promising actions may be mistakenly assigned negative advantages, leading RPEX to penalize them. Although the value functions in RPEX and other algorithms are trained on corrupted offline datasets, they still retain substantial information that is beneficial for online exploration. If the offline value functions are made more robust, as in RIQL, the value estimates become more reliable. As illustrated in Figures 2(d)–(e), when value functions are inaccurate, most penalized actions are suboptimal actions produced by the heavy-tailed policy. While IPW may also penalize some promising actions, it improves overall exploration efficiency (Figure 2(e)). This conclusion is further supported by RPEX’s superior performance in the non-corrupted setting. Moreover, such penalties can be tuned via $\kappa$ or the clipping range. In our experiments, we fix both $\kappa$ and the clipping range to demonstrate the inherent robustness of RPEX; fine-tuning these parameters could further improve exploration efficiency and performance.
>
> Across varying corruption levels, RPEX demonstrates strong robustness compared to RIQL-PEX, outperforming it by an average of approximately 16.8%, further validating the effectiveness of RPEX. We will include these experimental results in the revised version of our paper.
>
>
> > question regarding Eq. (6): Is $P_{w}$ clipped to zero when it takes negative values?
>
> We follow the official PEX implementation by using torch.distributions.Categorical with logit as input, which allows negative values, to implement Eq. (6). We do not clip negative values to zero, and our practical implementation strictly adheres to Eq. (6).
>
> > Typos in line 196-202
>
> Thank you for pointing this out. The correct reference is Eq. (6), not Eq. (5). We have corrected this typo in the revised version.

---

> > ### Comment · Reviewer_REEm · 2025-08-04
> > **No further questions**
> >
> > Thank you for the detailed and thoughtful rebuttal. I appreciate the authors'efforts in addressing my concerns. The clarifications and additional results provided have satisfactorily resolved the issues I raised. I have no further questions and am supportive of the paper.

---

> ### Author Response · Authors · 2025-08-04
> **Thank you for your support**
>
> Thank you for your support. We are pleased that our clarifications have addressed your concerns, and we sincerely appreciate your thoughtful feedback throughout the review process.

---

### Official Review · Reviewer_1RkU · 2025-06-30

**Clarity:** 3
**Significance:** 3
**Originality:** 3
**Rating:** 5
**Confidence:** 3

**Summary:**

This paper proposes an effective method, Robust Policy EXpansion (RPEX) for offline2online under the data corruption setting. The authors observe that data corruption induces heavy-tailed behavior in the policy, thereby substantially degrading the efficiency of online exploration. For this, they incorporate Inverse Probability Weighted (IPW) into the online exploration policy to alleviate heavy-tailedness. Both theoretical and empirical studies show the superiority of RPEX.

**Questions:**

Apart from the two points mentioned in Weaknesses, there are also some questions
1. I notice from Table 1 that RIQL, RIQL-PEX, and RPEX get very limited gain even negative gain in observation and dynamics attack. Can you provide some insights on this?
2. For the UTD ablation study, I think you can use another implementation way: only perform high UTD on critic but maintain standard (UTD=1) for actor. This is because in some literature, like REDQ [1], they only give the critic a high UTD rather than the actor. And in this case, maybe the issue you mentioned in Section 7.2 can be alleviated? I was wondering about the results in this case.

[1] Chen et al., "RANDOMIZED ENSEMBLED DOUBLE Q-LEARNING: LEARNING FAST WITHOUT A MODE", ICLR'21.

**Ethical Concerns:**

["NO or VERY MINOR ethics concerns only"]

**Final Justification:**

The authors test their methods on more tasks and run more seeds, which solves my major concern. So I decided to raise my score from 4 to 5.

**Limitations:**

The authors have discussed the limitations in their paper.

**Quality:**

3

**Strengths And Weaknesses:**

**Strengths**
1. It's novel that this work is the first one to propose a robust O2O rl algorithm.
2. The authors use quite clear toy examples to illustrate the problem of heavy-tailed behavior in the policy and how it influences online exploration.
3. Though this method introduces extra hyperparameters, the authors show that their method is not sensitive to them and use the same values across all tasks.

**Weaknesses**
1. I think the main experiment only considers too few tasks (HalfCheetah-MR, Walker2d-MR, Hopper-MR). It would be better to evaluate it on more benchmarks like AntMaze or Adroit.
2. All results are run on 5 seeds, which I think might not be that confident. From the training curves in the Appendix, it can be seen that on several tasks, the variances are very large. For these tasks, I think more runs can make the conclusion more convincing.

---

> ### Author Rebuttal · Authors · 2025-07-28
>
> We are grateful to the reviewer for their time and effort in reviewing our paper, as well as for the constructive suggestions. Below, we address your concerns one by one.
>
> > experiments on more tasks
>
> Thank you for pointing this out. We conducted additional experiments on the AntMaze-Large and AntMaze-Diverse tasks, which are the most challenging tasks in the AntMaze benchmark. We primarily focus on dynamics and observation corruption, as these represent the most difficult corruption settings.
>
> Following RIQL, we use a corruption range of 0.3 and a corruption rate of 0.2 during offline pretraining. For online corruption, we apply a corruption range of 0.5 and a corruption rate of 0.3. The offline value function is trained using RIQL with the default hyperparameters reported in the original RIQL paper. In the AntMaze experiments, we employ the same hyperparameters as those in the main paper (line 697) to demonstrate the robustness of RPEX. All results are averaged over 10 random seeds.
>
> The table below presents the performance comparison between RIQL-PEX and RPEX.
>
> |                           | RIQL-PEX            | RPEX                    |
> | ------------------------- | ------------------- | ----------------------- |
> | large-play dynamics       | 33.3->37.8$\pm$ 4.2 | **33.3->38.9$\pm$ 1.8** |
> | large-play observation    | 23.3->21.1$\pm$ 7.1 | **23.3->35.3$\pm$ 2.7** |
> | large-diverse dynamics    | 23.3->20.0$\pm$ 5.4 | **23.3->25.6$\pm$ 4.7** |
> | large-diverse observation | 26.7->25.6$\pm$ 8.1 | **26.7->36.7$\pm$ 5.3** |
> | Average                   | 26.6->26.1          | **26.6->34.1**          |
>
> As shown in the table above, RPEX outperforms RIQL-PEX by a large margin of approximately 30.7%.
> We will include these experiments in our paper.
>
> > more random seeds and large variance
>
> Thank you for pointing this out. As shown in lines 250–251, we use different corruption rates of 0.5 and 0.3 for the online and offline phases, respectively. Due to the high corruption probability, the training curves of algorithms may exhibit large variance. However, as shown in Table 1, RPEX achieves better performance with smaller variance compared to other methods. A similar high-variance phenomenon was also observed in RIQL [48], Table 1 (Hopper-MR reward: 84.8 $\pm$ 13.1), where results are averaged over 4 random seeds. To enhance the credibility of our findings, we conducted additional experiments on AntMaze-Large tasks under both observation and dynamics attacks using 10 random seeds. The results and experimental details are provided in the response to the concern titled “experiments on more tasks.”
>
> As demonstrated by the experimental results on the AntMaze tasks, RPEX achieves superior performance with lower variance compared to RIQL-PEX. These findings are consistent with those from experiments using five random seeds, further confirming the effectiveness of RPEX.
>
> > insights about limited gain in observation and dynamics attack
>
> Thank you for pointing this out. This is a very insightful question. As shown in Table 1 of our paper, all algorithms struggle to improve under observation and dynamics attacks. In particular, under observation attack, RIQL-PEX exhibits a performance drop of approximately 41% on the Hopper-MR task. Although RPEX significantly outperforms RIQL-PEX, it still struggles to improve performance. A similar phenomenon is also reported in RIQL [48].
>
> The primary cause of poor performance under observation attack is its strong disruption to online exploration. The key objective during the online phase is to collect optimal trajectories and update both the policy and the value function. When observation attacks occur, they affect action selection, potentially resulting in poor or even out-of-distribution (OOD) actions. Moreover, observation attacks can distort the Bellman backup, thereby hindering value function learning. According to Theorem 3 in RIQL [48], such suboptimal or OOD actions, combined with high Bellman backup error, lead to a loose upper bound on the difference between value functions learned under clean and corrupted observations. This explains why improving performance under observation attacks is particularly difficult.
>
> For dynamics attacks, the limited improvement after the O2O phase primarily stems from Bellman backup errors induced by the attack. Although RIQL reduces Bellman backup error by mitigating the heavy-tailedness of the Q-target, attacks on the dynamics during the online phase still distort the Bellman backup, thereby limiting performance improvement. We will include this discussion in the revised version of our paper.
>
> >  UTD only on critic
>
> Thank you for pointing this out. We conduct experiments in which only the critic is trained with a high update-to-data (UTD) ratio, while the actor maintains UTD = 1, on the Walker2d-MR tasks under dynamics and action corruption. All other hyperparameters remain the same as those used in our main paper (line 697). Following table shows the results of RPEX with such UTD implementation. The result for UTD=4 is averaged over 10 random seeds.
>
> | UTD | dynamics           | action                |
> | --- | ------------------ | --------------------- |
> | 1   | 80->92.2$\pm$ 1.4  | 85.9->118.9$\pm$ 10.1 |
> | 4   | 80->92.6$\pm$ 10.0 | 85.9->96.7$\pm$ 21.8  |
>
> As shown in the table above, the issue discussed in Section 7.2—namely, that high UTD primarily improves robustness against action corruption while slightly degrading overall performance—can be alleviated. However, we observe that although such UTD implementations can sometimes yield better performance, they also increase overall variance. This phenomenon may be attributed to the fact that data corruption introduces greater Bellman backup errors compared to the non-corrupted setting. While a high UTD ratio for the critic can accelerate learning, the subsequent Bellman backups may fluctuate significantly due to corrupted data. We will include these experiments in our revised paper.

---

> > ### Comment · Reviewer_1RkU · 2025-07-31
> > **Raising score**
> >
> > Thank the author's detailed response. I have no further questions, and raised my score to 5.

---

> ### Author Response · Authors · 2025-08-01
> **Thank you for raising the score!**
>
> Thank you for raising the score! Your constructive feedback has been instrumental in improving the quality of our work and we deeply appreciate your willingness to raise the score.

---

### Official Review · Reviewer_K7Qh · 2025-07-02

**Clarity:** 2
**Significance:** 2
**Originality:** 2
**Rating:** 3
**Confidence:** 4

**Summary:**

This paper addresses the challenge of robust offline-to-online reinforcement learning (O2O RL) under diverse data corruption, including noise and adversarial attacks on states, actions, rewards, and dynamics. To combat this, they introduce RPEX, which integrates Inverse Probability Weighting (IPW) into the Policy Expansion framework for adaptive action selection and improved exploration robustness. Theoretical justification is provided, and extensive experiments on D4RL benchmarks demonstrate that RPEX consistently outperforms strong baselines.

**Questions:**

How does RPEX fare in terms of computational overhead, wall-clock time, and scalability?

**Ethical Concerns:**

["NO or VERY MINOR ethics concerns only"]

**Final Justification:**

My view that the paper is incremental remains unchanged; nevertheless, in light of the substantial rebuttal material the authors have provided, I have opted to raise my score accordingly.

**Limitations:**

Please see weakness.

**Quality:**

3

**Strengths And Weaknesses:**

**Strengths**:

- The paper tackles an important and underexplored problem: robust O2O RL under both offline and online data corruption, which is central for real-world RL deployment.

- This paper has clear motivation, connecting the challenge of heavy-tailedness in policy distributions to degraded learning and exploration, and linking this to the inefficacy of prior robust RL approaches for O2O settings.

**Weaknesses**:
- The entire work is somewhat incremental, incorporating Inverse Probability Weighting—an established robustness tool from other domains—into PEX, which is designed for offline to online RL.

- Equation (6) is the most crucial part of the algorithm in this paper. However, some hyperparameters such as IPW Weight $\kappa$) and the range of Clip are not thoroughly discussed and analyzed in terms of their impact on performance. In other words, the ablation experiments are missing.

- The experimental environment is relatively limited, including only three GYM environments. Other more important environments, such as Maze and AntMaze, are not taken into consideration.

---

> ### Author Rebuttal · Authors · 2025-07-28
>
> We are grateful to the reviewer for their time and effort in reviewing our paper. Below, we address your concerns one by one.
>
> > How does RPEX fare in terms of computational overhead, wall-clock time, and scalability?
>
> We report the computational overhead in Appendix C.3, Table 2. In summary, RPEX improves performance by approximately 13.6% over PEX, with only an 8.1% increase in training time. These results suggest that our method enhances performance without incurring significant computational costs.
>
> Owing to its simplicity, RPEX can be integrated with a variety of offline RL and O2O RL methods (as also noted by Reviewer REEm). Our experiments demonstrate its integration with both RIQL and IQL, achieving a 13.6% performance improvement over PEX. These findings underscore that RPEX is both simple and effective.
>
>
> > Ablation study of IPW weight $\kappa$ and the range of clip
>
> Thank you for pointing this out.  We have conducted ablation study of IPW weight $\kappa$ in Fig 5. As shown in subfigure 3 in Fig 5, the performance gap of different $\kappa$ is relative small, the max gap is about 9%. Note that we use the same $\kappa=0.1$ and $\alpha^{-1}=3$ for our main results (Table 1), although further finetuning $\kappa$ can improve the performance according to the Fig 5. Such results demonstrate that RPEX is relatively insensitive to $\kappa$.
>
> For range of clip, we conduct ablation study on the Walker2d-MR environments under dynamics and action corruption. The reason to choose dynamics and action corruption is them represent two modular $s,a$ in the MDP. As shown in RIQL [48], dynamics corruption is known as difficulty for robust RL. The results of different clip range on the Walker2d-MR is shown in the following table. Note that the  $\kappa=0.1$,  $\alpha^{-1}=3$, corruption range (1) and rate (0.5) are as same as the Table 1. All results are averaged over 5 random seeds.
>
> | clip range     | dynamics           | action                 |
> | -------------- | ------------------ | ---------------------- |
> | (-10000,100)   | 80->92.2$\pm$ 1.4  | 85.9->118.9$\pm$ 10.1  |
> | (-100,100)     | 80->90.8$\pm$ 30.0 | 85.9->108.4$\pm$ 1.07  |
> | (0,100)        | 80->62.1$\pm$ 38.6 | 85.9->62.67$\pm$ 44.2  |
> | (-10000,10000) | 80->96.7$\pm$ 1.69 | 85.9->105.53$\pm$ 0.09 |
>
> As shown in the table above, the performance differences across various clipping ranges are minor, except for extreme cases such as the range (0, 100). Similar to $\kappa$, fine-tuning the clipping range may further improve performance. Throughout our paper, we consistently use the same values for $\kappa$ and the clipping range $(-10000, 100)$, which further demonstrates the robustness of RPEX. We will include these experiments in our paper.
>
> > More experimental environments
>
> Thank you for pointing this out. We conducted additional experiments on the AntMaze-Large and AntMaze-Diverse tasks, which are the most challenging tasks in the AntMaze benchmark. We primarily focus on dynamics and observation corruption, as these represent the most difficult corruption settings, as noted in both RIQL and our paper.
>
> Following RIQL, we use a corruption range of 0.3 and a corruption rate of 0.2 during offline pretraining. For online corruption, we apply a corruption range of 0.5 and a corruption rate of 0.3. The offline value function is trained using RIQL with the default hyperparameters reported in the original RIQL paper. In the AntMaze experiments, we employ the same hyperparameters as those in the main paper (line 697) to demonstrate the robustness of RPEX. All results are averaged over 10 random seeds.
>
> The table below presents the performance comparison between RIQL-PEX and RPEX.
>
> |                           | RIQL-PEX            | RPEX                    |
> | ------------------------- | ------------------- | ----------------------- |
> | large-play dynamics       | 33.3->37.8$\pm$ 4.2 | **33.3->38.9$\pm$ 1.8** |
> | large-play observation    | 23.3->21.1$\pm$ 7.1 | **23.3->35.3$\pm$ 2.7** |
> | large-diverse dynamics    | 23.3->20.0$\pm$ 5.4 | **23.3->25.6$\pm$ 4.7** |
> | large-diverse observation | 26.7->25.6$\pm$ 8.1 | **26.7->36.7$\pm$ 5.3** |
> | Average                   | 26.6->26.1          | **26.6->34.1**          |
>
> As shown in the table above, RPEX outperforms RIQL-PEX by a large margin of approximately 30.7%.
> We will include these experiments in our paper.
>
> >  The entire work is somewhat incremental
>
> Although IPW has been widely used in classification tasks, its role in reinforcement learning,  particularly in robust offline-to-online (O2O) RL, remains underexplored. To the best of our knowledge, RPEX is the first method to incorporate IPW into RL as a regularization term.
>
> Compared to PEX, RPEX introduces only minimal modifications yet achieves significantly better performance—improving upon PEX by approximately 13.6% with this simple adjustment. Furthermore, as noted by Reviewer REEm, RPEX is simple yet effective. Its lightweight modification to the PEX framework enables seamless integration with various offline RL algorithms while providing additional performance gains under both offline and online data corruption.
>
> In summary, the simplicity and scalability of RPEX are key advantages that enhance its practicality for real-world applications.

---

> > ### Comment · Area_Chair_E5Qv · 2025-08-06
> > **Could you reply to the authors' rebuttal?**
> >
> > Dear Reviewer K7Qh,
> >
> > Thanks a lot for reviewing this paper!
> >
> > When you have time, could you reply to the authors' rebuttal above? There are fewer than three days remaining in the author-reviewer discussion period.
> >
> > Thanks!

---

> > ### Comment · Reviewer_K7Qh · 2025-08-07
> >
> > I appreciate the authors for supplying the additional experimental details; incorporating them into the final manuscript will unquestionably make the paper more complete.
> >
> > Nevertheless, the overall contribution still reads like an “A + B” algorithm solving an “A + B” problem: IPW addresses robustness, while PEX handles the offline-to-online (O2O) RL, together yielding a robust O2O RL method. The authors’ response regarding “incremental” has not resolved my concern.

---

> ### Author Response · Authors · 2025-08-07
> **RPEX is simple and effective, rather than merely incremental.**
>
> Thank you for your response. As noted by reviewer REEm, “Investigating O2O RL under data corruption is an important and practically relevant direction, as such scenarios are common in real-world applications.” This underscores that O2O RL constitutes a significant and practical challenge for deploying RL in real-world settings [1,2], far beyond a simplistic “A+B” formulation.
>
> Moreover, our method is simple and effective, rather than merely incremental. The simplicity and scalability of RPEX are key advantages that enhance its practicality in real-world applications. As noted by reviewers 1RkU and REEm, identifying the cause that hinders O2O RL under data corruption is also one of our contributions. The reason we can achieve a performance gain of approximately 13.6% with minimal modification is that we demonstrate how data corruption not only amplifies the heavy-tailedness of Q-targets but also exacerbates that of the policy, which severely impairs the efficiency of online exploration (lines 46–48, Figure 2, lines 167-186). IPW is employed to mitigate the heavy-tailedness of the policy, thereby improving online exploration efficiency. This enhanced efficiency leads to better actions and states being selected and used to train the policy. Consequently, IPW contributes to improved O2O performance under data corruption. Notably, the original use of IPW is to address heavy-tailed classification problems [3,4], rather than enhancing robustness under data corruption. Our motivation for incorporating IPW into O2O RL arises from the aforementioned insight, rather than the oversimplified view that “IPW addresses robustness, while PEX handles the offline-to-online (O2O) RL”.
>
>
> In summary, our contributions are as follows:
>
> 1. We show that data corruption not only amplifies the heavy-tailedness of Q-targets but also exacerbates the heavy-tailedness of the policy, significantly impairing the efficiency of online exploration.
>
> 2. We propose RPEX, a simple yet effective O2O reinforcement learning algorithm that incorporates IPW into PEX to address the O2O problem under data corruption.
>
> 3. To the best of our knowledge, RPEX is the first robust O2O reinforcement learning algorithm that integrates IPW into RL, achieving efficient and robust performance improvements under both offline and online data corruption.
>
> 4. Our theoretical analysis supports the effectiveness of our approach, demonstrating that applying IPW mitigates the heavy-tailedness induced by data corruption and facilitates efficient exploration. With minimal modifications, extensive experiments show that RPEX improves performance by approximately 13.6% over baseline methods. We also conduct comprehensive ablation studies to assess the effectiveness of common O2O techniques, such as maintaining the offline buffer and increasing the update-to-data (UTD) ratio.
>
> [1] Jiawei Xu, Rui Yang, Shuang Qiu, Feng Luo, Meng Fang, Baoxiang Wang, and Lei Han. Tackling data corruption in offline reinforcement learning via sequence modeling,
>
> [2] Rui Yang, Han Zhong, Jiawei Xu, Amy Zhang, Chongjie Zhang, Lei Han, and Tong Zhang. Towards robust offline reinforcement learning under diverse data corruption.
>
> [3] Yin Cui, Menglin Jia, Tsung-Yi Lin, Yang Song, and Serge Belongie. Class-balanced loss based on effective number of samples.
>
> [4] Chongsheng Zhang, George Almpanidis, Gaojuan Fan, Binquan Deng, Yanbo Zhang, Ji Liu, Aouaidjia Kamel, Paolo Soda, and João Gama. A systematic review on long-tailed learning

---

> > ### Comment · Reviewer_K7Qh · 2025-08-09
> >
> > Thank you for the authors’ thorough response and continued discussion.
> >
> > Frankly, I still regard the work as incremental in both problem framing and methodology. Nevertheless, the authors’ incisive analysis of the underlying issue and their well-argued motivation for adopting IPW are commendable. Coupled with the extensive additional experiments provided during the rebuttal, I am raising my score to 3 to reflect my neutral stance. Whether the paper should be accepted is a decision I leave to the other reviewers and the AC.

---

> > > ### Author Response · Authors · 2025-08-09
> > > **Thank you for increasing the score**
> > >
> > > Thank you for increasing the score. We will incorporate the clarifications regarding our contributions and novelties, as well as the additional experiments discussed during the rebuttal, into the final version of our paper.

---

> ### Author Response · Authors · 2025-08-09
> **Sincerely Looking Forward to Your Feedback**
>
> Dear Reviewer K7Qh,
>
> We would like to sincerely thank you once again for your valuable feedback, insightful comments, and the time and effort you’ve dedicated to reviewing our work. Your constructive suggestions have been immensely helpful in improving our paper.
>
> As the discussion period is coming to a close in **less than 9 hours**, we wanted to kindly check if our response has fully addressed your concerns. If there are any remaining questions or points for discussion, we would be more than happy to engage further.
>
> If our response has resolved your concerns, we would be truly grateful if you might consider increasing your support. Of course, we fully respect your decision either way and deeply appreciate your thoughtful review. Your support would mean a great deal to us!
>
> Best regards,
>
> The authors of Submission 1198

---

### Official Review · Reviewer_moaS · 2025-07-03

**Clarity:** 2
**Significance:** 2
**Originality:** 2
**Rating:** 4
**Confidence:** 3

**Summary:**

This paper proposes a method for learning an efficient and robust policy that can handle both offline and online data corruption, aiming to prevent unsafe behavior in real-world environments. The paper prove the effectiveness of their approach through extensive experiments across various data corruption settings on D4RL.

**Questions:**

Please see Weaknesses part. And in my opinion, it would be helpful to clarify whether the theoretical derivation can provide an explanation for the role of IPW in addressing data corruption.

**Ethical Concerns:**

["NO or VERY MINOR ethics concerns only"]

**Final Justification:**

The authors’ explanations regarding the theoretical derivation, as well as their discussion on several issues raised by other reviewers, have addressed part of my concerns. I am therefore considering increasing my score by 1 point. However, I still remain concerned about the novelty of the work, particularly in comparison with PEX.

**Limitations:**

See Weaknesses.

**Quality:**

2

**Strengths And Weaknesses:**

**Strengths**:
1. The authors demonstrate the effectiveness of their IPW-corrected PEX method under data corruption through extensive experiments. The experimental evaluation is thorough and well-designed across multiple data corruption settings.
2. The proposed method is conceptually simple yet practically effective.

**Weaknesses**:
The proposed method lacks sufficient theoretical justification. The mathematical derivation presented in the paper largely builds upon prior work [48, 51], and the core algorithm shows limited novelty especially when compared with [51]. Specifically, in Appendix B1, while the authors explain how the key Equation (6) is obtained, the subsequent steps from (17) to (19) are not sufficiently justified which weakens the theoretical foundation of the method.

---

> ### Author Rebuttal · Authors · 2025-07-28
>
> We are grateful to the reviewer for their time and effort in reviewing our paper. Below, we address your concerns one by one.
>
> > limited novelty
>
> First, as stated in the problem statement, our motivation is to develop a robust algorithm capable of handling both offline and online data corruption. This motivation differs from that of RIQL [48], which focuses on robust offline learning, and PEX, which aims to develop an efficient offline-to-online (O2O) method. Addressing both offline and online corruption is critical for real-world RL applications. To the best of our knowledge, our method is the first to propose a robust O2O reinforcement learning algorithm. We also conduct extensive ablation studies to analyze the effectiveness of commonly used O2O techniques, such as retaining the offline buffer and increasing the update-to-data (UTD) ratio.
>
> Second, compared to PEX [51], RPEX (ours) incorporates inverse probability weighting (IPW) into the action selection weighting (Eq. 6). Although simple, our toy experiment (Fig. 2) demonstrates that this modification mitigates the heavy-tailed behavior of the policy, thereby enhancing exploration efficiency. Extensive experiments show that this modification improves performance by approximately 13.6% over vanilla PEX. Beyond integration with RIQL, RPEX is also compatible with IQL and outperforms IQL+PEX by about 12.4% under both offline and online corruption. The simplicity and scalability of RPEX enable seamless integration with various offline RL algorithms, making it a promising approach for future robust O2O research (as also noted by Reviewer REEm).
>
> Third, although IPW is commonly used in classification tasks, to the best of our knowledge, our work is the first to apply IPW in reinforcement learning and demonstrate its effectiveness.
>
> In summary, while RPEX can be integrated with RIQL, it is also applicable to other offline RL methods such as IQL. Although Eq. 6 resembles that of PEX, the motivation behind RPEX is distinct. Our experiments demonstrate that this simple modification significantly enhances the robustness of PEX. The simplicity and scalability of RPEX are key advantages that support its practicality in real-world applications.
>
> > lack of theoretical justification
>
> As shown in Proposition 6.1, Eq. (6) is derived from Eq. (9), which seeks to maximize both the expected reward and entropy, while incorporating a regularization term $\frac{(Q - V)}{\pi_{i}(a_{i}|s)}P_{w}(i|a_{1}, a_{2})$. This regularization term consists of  IPW multiplied by the composite policy $P_w$. The inclusion of $P_w$ enables the computation of its gradient, thereby mitigating the heavy-tailedness of the composite policy. We further elaborate on the role of this regularization in lines 231–239.
>
> In essence, the regularization term reduces the value of $P_w$ for suboptimal actions (i.e., those with $Q - V < 0$), particularly when such actions are taken with low probability (as captured by the IPW term). This indicates that these actions are likely produced by the policy due to heavy-tailedness induced by data corruption. The IPW component serves to suppress these corrupted actions while promoting better ones, thereby enhancing exploration efficiency.
>
> The above analysis illustrates the rationale behind this regularization term and supports Eqs. (17)–(19). The theoretical derivation, along with the intuitive explanation in lines 196–204, clarifies the role of IPW in alleviating policy heavy-tailedness and improving exploration efficiency under data corruption.

---

> > ### Comment · Area_Chair_E5Qv · 2025-08-06
> > **Might you respond to the author rebuttal**
> >
> > Dear Reviewer moaS,
> >
> > Thanks a lot for reviewing this paper!
> >
> > When you have time, could you reply to the authors' rebuttal above? There are fewer than three days remaining in the author-reviewer discussion period.
> >
> > Thanks!

---

> ### Comment · Reviewer_moaS · 2025-08-08
>
> The authors’ explanations regarding the theoretical derivation, as well as their discussion on several issues raised by other reviewers, have addressed part of my concerns. I am therefore considering increasing my score by 1 point. However, I still remain concerned about the novelty of the work, particularly in comparison with PEX.

---

> ### Author Response · Authors · 2025-08-08
> **Thank you for your response and for your willingness to increase the score**
>
> Thank you for your response and for your willingness to increase the score. We greatly appreciate your constructive feedback, which has been instrumental in enhancing the quality of our work. Here, we further clarify the differences and novelty of our proposed method, RPEX, in comparison with PEX.
>
> 1. As stated in lines 88–90, RPEX focuses on developing a robust O2O algorithm capable of handling various types of data corruption in both offline datasets and online transitions, whereas PEX [1] addresses the unlearning phenomenon arising from directly transferring an offline policy to the online phase. As noted by reviewer REEm, “Investigating O2O RL under data corruption is an important and practically relevant direction, as such scenarios are common in real-world applications.” To the best of our knowledge, RPEX is the first robust O2O reinforcement learning algorithm to integrate IPW into RL, achieving substantial and robust performance improvements under both offline and online data corruption. This distinction highlights both the methodological differences and the novelty of RPEX.
>
> 2. The key novelty of RPEX lies in our observation that data corruption not only amplifies the heavy-tailedness of Q-targets but also exacerbates the heavy-tailedness of the policy. Building on this insight, we incorporate IPW into the PEX framework, yielding significant performance gains over vanilla PEX. As emphasized by reviewers 1RkU and REEm, identifying the underlying causes that hinder O2O RL under data corruption is itself an important contribution and a core novelty of our work. Furthermore, due to its simplicity, RPEX is easy to implement and practical for real-world applications.
>
> 3. In our practical implementation, we additionally employ well-established O2O techniques such as UTD (updates-to-data) [2] to examine its influence in the presence of adversarial attacks (lines 273-280), whereas PEX does not adopt UTD. This represents another clear difference between RPEX and PEX.
>
> We will include a more detailed discussion of the differences and novelty between our method and PEX in the final revision.
>
> [1] Zhang, H., Xu, W., Yu, H., 2023. Policy Expansion for Bridging Offline-to-Online Reinforcement Learning.
>
> [2] Zhou, Z., Peng, A., Li, Q., Levine, S., Kumar, A., 2025. EFFICIENT ONLINE REINFORCEMENT LEARNING FINE-TUNING NEED NOT RETAIN OFFLINE DATA.

---

### Note · Authors · 2025-08-11

We sincerely thank the AC and all reviewers for their time and valuable feedback on our submission. The primary remaining concern pertains to the novelty of our work, given the simplicity of our method. However, we contend that simplicity is one of RPEX’s greatest strengths, as it makes the method easy to implement and practical for real-world applications, particularly in corruption scenarios. Furthermore, RPEX employs the same parameter values across all tasks; this insensitivity is partly attributable to its simplicity. More importantly, RPEX’s performance improvement over PEX with minimal modification stems from the insight that data corruption not only intensifies the heavy-tailedness of Q-targets but also aggravates that of the policy, thereby significantly reducing the efficiency of online exploration. Building on this observation, we integrate IPW into PEX to mitigate the heavy-tailedness of the policy and enhance exploration efficiency, thereby proposing RPEX—a simple yet effective approach to addressing the O2O RL problem under data corruption. All reviewers have acknowledged this contribution as one of our key novelties. In summary, our main contributions and novelties are:

1. We propose RPEX, a simple yet effective O2O reinforcement learning algorithm that incorporates IPW into PEX to address the O2O problem under data corruption. To the best of our knowledge, RPEX is the first reinforcement learning method to integrate IPW into RL.

2. We demonstrate that data corruption not only amplifies the heavy-tailedness of Q-targets but also exacerbates the heavy-tailedness of the policy, thereby substantially impairing the efficiency of online exploration.

3. To the best of our knowledge, RPEX is the first robust O2O reinforcement learning algorithm that achieves efficient and reliable performance improvements under both offline and online data corruption.

4. Our theoretical analysis supports the effectiveness of our approach, demonstrating that applying IPW mitigates the heavy-tailedness induced by data corruption and facilitates efficient exploration. With minimal modifications, extensive experiments show that RPEX improves performance by approximately 13.6% over baseline methods. We also conduct comprehensive ablation studies to assess the effectiveness of common O2O techniques, such as maintaining the offline buffer and increasing the update-to-data (UTD) ratio.

---

### Decision · Program_Chairs · 2025-09-17

**Decision:**

Accept (poster)

**Comment:**

This paper proposes an effective method, Robust Policy EXpansion (RPEX), for offline-to-online reinforcement learning under the data corruption setting. It observes that data corruption induces heavy-tailed behavior in the policy, thereby substantially degrading the efficiency of online exploration. To resolve this, this paper incorporates Inverse Probability Weighted (IPW) into the online exploration policy to alleviate heavy-tailedness. Experiment results have shown the good performance of RPEX.

This paper is borderline and probably more on the acceptance side. For the strengths, to the best of our knowledge, this paper is the first to propose a robust offline-to-online RL algorithm. It also uses clear toy examples to illustrate the problem of heavy-tailed behavior in the policy and discuss how it influences online exploration. The proposed RPEX algorithm is simple and scalable. With the additional experimental results mentioned in the authors' rebuttals, the experimental results of this paper also look strong.

However, on the weakness side, though this paper proposes a novel algorithm for a novel problem, to experts in this field, both the considered problem (robust + offline-to-online RL) and the proposed algorithm (PEX + IPW) might seem a bit incremental. In other words, both the problem formulation and the developed algorithm might not be surprising to experts. This limits the novelty of this paper.

Overall, I think this is a borderline paper, slightly more on the acceptance side. I recommend accepting this paper.